# Disordered $\mathcal{N} = (2, 2)$ Supersymmetric Field Theories

Chi-Ming Chang[a,b], Xiaoyang Shen[c,d]

[a]*Yau Mathematical Sciences Center (YMSC), Tsinghua University, Beijing 100084, China*

[b]*Beijing Institute of Mathematical Sciences and Applications (BIMSA), Beijing 101408, China*

[c]*Department of Physics, Tsinghua University, Beijing 100084, China*

[d]*Institute for Advanced Study, Tsinghua University, Beijing 100084, China*

cmchang@tsinghua.edu.cn   xiaoyangshum@gmail.com

## Abstract

We investigate a large class of $\mathcal{N} = (2, 2)$ supersymmetric field theories in two dimensions, which contains the Murugan-Stanford-Witten model, and can be naturally regarded as a disordered generalization of the two-dimensional Landau-Ginzburg models. We analyze the two and four-point functions of chiral superfields, and extract from them the central charge, the operator spectrum, and the chaos exponent in these models. Some of the models exhibit a conformal manifold parameterized by the variances of the random couplings. We compute the Zamolodchikov metrics on the conformal manifold, and demonstrate that the chaos exponent varies nontrivally along the conformal manifolds. Finally, we introduce and perform some preliminary analysis of a disordered generalization of the gauged linear sigma models, and discuss the low energy theories as ensemble averages of Calabi-Yau sigma models over complex structure moduli space.

# 1 Introduction

Disordered couplings have provided us a large class of large $N$ solvable models, and brought many new insights into the dynamics of black holes in quantum gravity. The classic example is the Sachdev-Ye-Kitaev (SYK) model [1, 2], which is a quantum mechanical system of $N$ Majorana fermions interacting with random multi-fermion couplings. Using large $N$ techniques, the correlation functions of the fermions in the SYK model can be explicitly solved [3–5]. For instance, the two-point function can be solved by summing over the melonic diagrams using the Schwinger-Dyson equation, and the four-point function is solved by

summing over the ladder diagrams. Interesting physical observables are then extracted from these exact solutions, such as the spectrum of two-particle states and the chaos exponent from the Euclidean and the out-of-time order four-point correlation functions, respectively. They reveal many remarkable properties of the SYK model.

At low temperatures, the SYK model exhibits an emergent time reparameterization symmetry, which is weakly broken by finite temperature leading to a Goldstone mode, the Schwarzian sector [4, 6]. Despite the low energy spectrum of the SYK model is not sparse, the Schwarzian sector dominates over the rest of the states. Consequently, the holographic dual at low energies is governed by a two-dimensional dilaton gravity, the Jackiw Teitelboim (JT) gravity [6–8]. The SYK model further displays maximal chaos, as the chaos exponent saturates the bound on chaos [9], which is a notable feature that shares with black holes in Einstein gravity [10].

Over the years, the SYK model has been generalized to include complex fermions [11–13], additional flavor symmetry [14], and supersymmetry [15–20]. Going beyond 0+1 dimensions, the two and three-dimensional generalizations of the SYK have been studied with various numbers of supersymmetries [21–27]. In higher dimensions, one has to consider nontrivial renormalization group (RG) flows, which introduce additional complications. On the one hand, the couplings involving only fermions are (marginally) irrelevant in two dimensions and above. On the other hand, the bosonic models typically require fine-tunings of the relevant couplings to reach the conformal fixed point in the infrared (IR), which becomes subtle when the couplings are random variables. Nevertheless, with $\mathcal{N} = 2$ supersymmetry in two dimensions, the Murugan-Stanford-Witten (MSW) model, introduced in [21], overcomes both problems and admits a superconformal fixed point.

In this paper, we study generalizations of the MSW model by introducing multiple families of disordered chiral superfields. The models are solvable in the large $N$ limit, defined as the numbers of the chiral superfields in each family becoming large while the ratios between the numbers remain finite. They can also be viewed as the disordered generalization of the $\mathcal{N} = 2$ Landau-Ginzburg models in two dimensions, and follow a similar classification [28–30] (see Section 2.2).[1] The MSW model is the simplest model in the classification with only one family of chiral superfields. An important new feature of the more general disordered Landau-Ginzburg models is that when there are two or more families of chiral superfields, the models could admit nontrivial conformal manifolds in the IR parametrizing by the variances of the random couplings. We investigate several examples in the classification with two families of chiral superfields, including one with an IR conformal manifold (see Section 2.4). In particular, we compute the two and four-point functions of the chiral superfields in these models in the large $N$ limit by summing over the melonic and ladder diagrams, and we

---

[1]A closely related tensor model generalization of the $\mathcal{N} = 2$ Landau-Ginzburg models was studied in [31].

extract the chaos exponents from the four-point functions. In general, the chaos exponent $\lambda_L$ depends on the ratio of the numbers of chiral superfields in each family, as well as the coordinates of the conformal manifold (when the manifold exists). We find an upper bound $\lambda_L \lesssim 0.5824$ across all the examples we studied. We propose that this is a universal upper bound for the chaos exponents in the disordered Landau-Ginzburg models.

Besides large $N$ techniques, the disordered Landau-Ginzburg models can also be studied by supersymmetric localization. Following the analysis of the non-disordered models in [32–34], we compute the two-sphere partition functions and the two-point functions of the disordered models (see Section 2.5). In the large $N$ limit, the results of the two-point function coefficients agree nicely with those computed before from summing Feynman diagrams. This provides extra evidence that the disordered Landau-Ginzburg models flow to superconformal fixed points in the IR. Furthermore, in the example with an IR conformal manifold, we compute the Zomoldchikov metric by taking derivatives of the two-sphere partition function.

Another new feature when there are multiple families of chiral superfields is that the superpotential could be engineered such that the theory possesses nontrivial flavor U(1) symmetries. Such a superpotential always has flat directions, and the IR theory is non-compact. One could make the theory compact by gauging the U(1) flavor symmetries, where the $D$-terms potential lifts all the flat directions. The resulting theory is a disordered generalization of the gauged linear sigma models. In the seminal work [35], it was shown that the (non-disordered) gauged linear sigma models, with an anomalous-free axial R-symmetry and a positive Fayet-Iliopoulos coupling, are in the same universality class as the nonlinear sigma models on Calabi-Yau target spaces, i.e. they flow to the same $\mathcal{N} = (2,2)$ superconformal field theories. This result implies that the disordered gauged linear sigma models, with the same conditions as above, are IR-dual to the ensemble averages of the Calabi-Yau sigma models over the complex structure moduli (see Section 3). To support this, we compute the two-point functions of the chiral superfields and the result confirms that the theories flow to IR superconformal fixed points.

The remainder of this paper is organized as follows. Section 2.2 introduces the disordered Landau-Ginzburg models and presents a classification of the models. Section 2.3 reviews the Murugan-Stanford-Witten model. Section 2.4 studies examples of the disordered Landau-Ginzburg models with two families of chiral superfields, computing the two and four-point functions and analyzing the chaos exponents. Section 2.5 applies the supersymmetric localization to the disordered Landau-Ginzburg models, and computes the two-sphere partition functions, two-point functions, and the Zamolodchikov metric for several examples. Section 3 introduces the disordered gauged linear sigma models, discusses their relations to the ensemble averages of Calabi-Yau sigma models, and performs some preliminary analysis.

# 2 Disordered Landau-Ginzburg models

## 2.1 The models

Let us consider a disordered $\mathcal{N} = 2$ Ginzburg-Landau model with $n$ different families of chiral superfields: $\Phi_i^{(1)}$ for $i = 1, \cdots, N_1$, $\Phi_i^{(2)}$ for $i = 1, \cdots, N_2$, and so on. The chiral superfields have a standard kinetic term

$$\mathcal{L}_{\text{kin}} = \int d^2\theta d^2\tilde{\theta} \left( \widetilde{\Phi}^{(1),i}\Phi_i^{(1)} + \widetilde{\Phi}^{(2),i}\Phi_i^{(2)} + \cdots + \widetilde{\Phi}^{(n),i}\Phi_i^{(n)} \right) , \tag{2.1}$$

and are coupled via an interaction term

$$\mathcal{L}_W = i \int d^2\theta \, W(\Phi_i^{(1)}, \cdots, \Phi_i^{(n)})\Big|_{\tilde{\theta}=\bar{\tilde{\theta}}=0} + \text{h.c.} . \tag{2.2}$$

Our conventions of the superspace are given in Appendix A. The disordered superpotential $W$ contains terms with random couplings, with the general form as

$$W = \sum_{p\equiv(p_1,\cdots,p_n)\in\mathcal{I}} g_p^{I_1\cdots I_n}(\Phi_{I_1}^{(1)})^{p_1} \cdots (\Phi_{I_n}^{(n)})^{p_n} , \tag{2.3}$$

where $\mathcal{I}$ is an index set that controls which terms would appear in the superpotential, the index $I_p$ is a collection of $p_a$ indices, $I_a = (i_1, \cdots, i_{p_a})$, and $(\Phi_{I_a}^{(a)})^{p_a}$ stands for

$$(\Phi_I^{(a)})^p \equiv \Phi_{i_1}^{(a)} \cdots \Phi_{i_p}^{(a)} . \tag{2.4}$$

The coupling constants $g_p^{I_1\cdots I_n}$ are independent Gaussian random variables with zero mean and variance as

$$\left\langle g_p^{I_1\cdots I_n} \right\rangle = 0 , \quad \left\langle g_p^{I_1\cdots I_n}\overline{g}_{p,I_1'\cdots I_n'} \right\rangle = \frac{J_p^2}{N^{p_1+\cdots+p_n-1}}\delta_{I_1'}^{I_1}\cdots\delta_{I_n'}^{I_n} , \quad \delta_{I'}^I \equiv \delta_{i_1'}^{(i_1}\cdots\delta_{i_p'}^{i_p)} , \tag{2.5}$$

where $N = N_1 + \cdots + N_n$, We are interested in the limit $N_i \to \infty$ while fixing $J_p$ and the ratios

$$\lambda_i = \frac{N_i}{N} . \tag{2.6}$$

The superspace coordinates $\theta^+$ and $\theta^-$ have charges $(1,0)$ and $(0,1)$ under $\text{U}(1)_L \times \text{U}(1)_R$ R-symmetry, and the coordinates $\bar{\theta}^{\pm}$ have the opposite charges. For the interaction terms to preserve the $\text{U}(1)_L \times \text{U}(1)_R$ symmetry, the superpotential has to be a quasi-homogeneous polynomial, and we further demand that the chiral superfields in the same family scale by the same weight, i.e.

$$W(\lambda^{q_1}\Phi_i^{(1)}, \cdots, \lambda^{q_n}\Phi_i^{(n)}) = \lambda W(\Phi_i^{(1)}, \cdots, \Phi_i^{(n)}) \quad \text{for} \quad \lambda \in \mathbb{C}^* . \tag{2.7}$$

Under the renormalization group, the theory flows to a strongly coupled $\mathcal{N} = (2,2)$ SCFT. The $U(1)_L \times U(1)_R$ R-symmetry becomes the part of the superconformal algebra. The bottom component of the chiral superfields $\Phi^{(a)}$ become chiral primary operators of R-charges $(q_a, q_a)$. By the quasi-homogeneity condition, the powers $p_a$ in (2.3) and the R-charges $q_a$ satisfy

$$\sum_{a=1}^{n} p_a q_a = 1 \,. \tag{2.8}$$

For a given $(p_1, \cdots, p_n)$, we focus on the cases that the couplings $g_p^{I_1, \cdots, I_n}$ are generic, since generic couplings give dominant contributions to the ensemble average over the coupling constants.

**IR conformal manifold and field redefinitionss** In the non-disordered models, the coefficients in the superpotential modulo (quasi-homogeneous) field redefinitions of the chiral superfields correspond to exactly marginal deformations of the IR SCFTs. In disordered models, the coefficients in the superpotential are random couplings and should be averaged over, but we could still vary the variances $J_p$ in (2.5). Some of the variances could be fixed again by field redefinitions (of bilocal superfields), and the remaining variances give marginal deformations and parameterize the IR conformal manifold of the disordered models.

To see more precisely how field redefinitions fix the variances, let us integrate out the random couplings $g_p^{I_1 \cdots I_n}$'s and arrive at the action of the bilocal superfields $G^{(a)}(\widetilde{Z}_1, Z_2)$ and $\Sigma^{(a)}(\widetilde{Z}_1, Z_2)$,

$$
\begin{aligned}
S = &\sum_a N_a \log \det \left[ \theta_{12} \bar{\bar{\theta}}_{12} \delta(\langle 12 \rangle) \delta(\langle \bar{1}\bar{2} \rangle) D_2 \overline{D}_2 + \Sigma^{(a)}(\widetilde{Z}_1, Z_2) \right] + N \sum_a \mathrm{Tr}\left( \Sigma^{(a)} \cdot G^{(a)} \right) \\
&- N \int d^{2|2} \widetilde{Z}_1 d^{2|2} Z_2 \sum_{p \equiv (p_1, \cdots, p_n) \in \mathcal{I}} J_p^2 G^{(1)}(\widetilde{Z}_1, Z_2)^{p_1} \cdots G^{(n)}(\widetilde{Z}_1, Z_2)^{p_n} \,,
\end{aligned} \tag{2.9}
$$

where $Z = (y, \bar{y}, \theta, \bar{\theta})$, $\widetilde{Z} = (\tilde{y}, \bar{\tilde{y}}, \tilde{\theta}, \bar{\tilde{\theta}})$. The super-derivatives $D, \overline{D}$ are defined in A.2, and the super-distances $\langle 12 \rangle$ and $\langle \bar{1}\bar{2} \rangle$ are defined in (A.9). We have used the matrix notation for the second term on the first line of (2.9) as

$$\mathrm{Tr}\left( \Sigma^{(a)} \cdot G^{(a)} \right) = \int d^{2|2} \widetilde{Z}_1 d^{2|2} Z_2 \, \Sigma^{(a)}(\widetilde{Z}_1, Z_2) G^{(a)}(\widetilde{Z}_1, Z_2) \,. \tag{2.10}$$

In the low energy limit $E \ll J_p$, we can drop the derivative term $D_2 \overline{D}_2$ in (2.9).

Now, we follow the arguments in [36,37] (with suitable generalizations to bilocal actions) to show that one could use field redefinitions to simplify the action (2.9). Consider the field redefinition of the bilocal field $G^{(a)}$ as

$$G^{(a)}(\widetilde{Z}_1, Z_2) \to G^{(a)\prime}(\widetilde{Z}_1, Z_2) = F^{(a)}(G^{(1)}(\widetilde{Z}_1, Z_2), \cdots, G^{(n)}(\widetilde{Z}_1, Z_2)) \,, \tag{2.11}$$

where $F^{(a)}$ is a quasi-homogeneous polynomial that has the same homogeneous degree as $G^{(a)}$, i.e.

$$F^{(a)}\big(\lambda^{q_1}G^{(1)}(\widetilde{Z}_1, Z_2), \cdots, \lambda^{q_n}G^{(n)}(\widetilde{Z}_1, Z_2)\big) = \lambda^{q_a}F^{(a)}\big(G^{(1)}(\widetilde{Z}_1, Z_2), \cdots, G^{(n)}(\widetilde{Z}_1, Z_2)\big). \quad (2.12)$$

Under the field redefinition (2.11), the path integral measure $DG^{(a)}$ becomes

$$DG^{(a)} \to DG^{(a)'} = |\det(\delta F/\delta G)|DG^{(a)}. \quad (2.13)$$

The Jacobian $|\det(\delta F/\delta G)|$ is a constant and can be ignored because $\delta F/\delta G$ could be arranged to a block upper triangular matrix with constant diagonal blocks by the quasi-homogeneous condition (2.12).

Next, we consider the field redefinition of $\Sigma^{(a)}$ as

$$\Sigma^{(a)} \to \Sigma^{(a)'} = \Sigma^{(a)} \cdot G^{(a)} \cdot (F^{(a)})^{-1}, \quad (2.14)$$

where "$\cdot$" is the matrix product that stands for integrating over $Z$ or $\widetilde{Z}$ as in (2.10), and $(F^{(a)})^{-1}$ is the matrix inverse of $F^{(a)}$. Under the field redefinition (2.14), the path integration measure $D\Sigma^{(a)}$ changes to

$$D\Sigma^{(a)} \to D\Sigma^{(a)'} = \left|\det\left[G^{(a)} \cdot (F^{(a)})^{-1}\right]\right|^V D\Sigma^{(a)}, \quad (2.15)$$

where $V$ is the rank of the bilocal superfield $\Sigma^{(a)}$ regarded as a matrix.[2] In summary, we arrive at the action

$$\begin{aligned}
S = &\sum_a N_a \log\det\left(\Sigma^{(a)}\right) + N\sum_a \text{Tr}\left(\Sigma^{(a)} \cdot G^{(a)}\right) \\
&- N\int d^{2|2}\widetilde{Z}_1 d^{2|2}Z_2 \sum_{p\equiv(p_1,\cdots,p_n)\in\mathcal{I}} J_p^2 F^{(1)}(\widetilde{Z}_1, Z_2)^{p_1}\cdots F^{(n)}(\widetilde{Z}_1, Z_2)^{p_n} \\
&- \sum_a (N_a - V)\log\det\left(F^{(a)} \cdot (G^{(a)})^{-1}\right).
\end{aligned} \quad (2.16)$$

The last term in (2.16) can be written as a ghost action

$$S_{\text{gh}} = \sum_a (N_a - V)\int d^{2|2}\widetilde{Z}_1 d^{2|2}Z_2\, \widetilde{C}(\widetilde{Z}_1)\left[F^{(a)} \cdot (G^{(a)})^{-1}\right](\widetilde{Z}_1, Z_2)C(Z_2), \quad (2.17)$$

where $\widetilde{C}$ and $C$ are the anti-chiral and chiral ghost superfields, respectively. Because $F^{(a)}$ is a quasi-homogeneous polynomial with the same degree as $G^{(a)}$, it should take the form as

$$F^{(a)}(G^{(1)}, \cdots, G^{(n)}) = \kappa G^{(a)} + H^{(a)}(G^{(1)}, \cdots, \cancel{G^{(a)}}, \cdots, G^{(n)}), \quad (2.18)$$

___
[2]More precise definition of $V$ can be found in [38].

where $H^{(a)}$ is a quasi-homogeneous polynomial that does not depend on $G^{(a)}$. We have

$$\left[F^{(a)} \cdot (G^{(a)})^{-1}\right](\widetilde{Z}_1, Z_2) = \kappa \theta_{12} \bar{\bar{\theta}}_{12} \delta(\langle 12 \rangle) \delta(\langle \bar{1}\bar{2} \rangle) + \left[H^{(a)} \cdot (G^{(a)})^{-1}\right](\widetilde{Z}_1, Z_2). \qquad (2.19)$$

Substituting (2.19) into the ghost action (2.17), the first term in (2.19) gives a mass term for the ghost fields $C$ and $\widetilde{C}$. Hence, in the IR limit, we can integrate out the ghost fields $C$ and $\widetilde{C}$, equivalent to deleting the last term in (2.16). Because $F^{(a)}$'s are quasi-homogeneous, the second line of (2.16) can be rewritten as

$$-N \int d^{2|2}\widetilde{Z}_1 d^{2|2} Z_2 \sum_{p \equiv (p_1, \cdots, p_n) \in \mathcal{I}} J_p'^2 G^{(1)}(\widetilde{Z}_1, Z_2)^{p_1} \cdots G^{(n)}(\widetilde{Z}_1, Z_2)^{p_n}, \qquad (2.20)$$

which takes the same form as the second line of (2.9), but with new coefficients $J_p'^2$ which are linear combinations of the old coefficients $J_p^2$. Hence, the field redefinition (2.11) gives us equivalence relations between variances

$$J_p'^2 \sim J_p^2, \qquad (2.21)$$

which can be used to fix (some of) the variances $J_p^2$.

## 2.2 A classification

We presently discuss the constraints and classifications of the disordered superpotential $W$. In this discussion, we could treat a family of superfields $\{\Phi_i^{(a)} \mid i = 1, \cdots, N_a\}$ as a single superfield $\Phi^{(a)}$, and treat the superpotential $W$ as a function of the variables $\Phi^{(1)}, \cdots, \Phi^{(n)}$. The classification problem now reduces to the problem of classifying non-disordered Ginzburg-Landau theories ($N_i = 1$ for all $i = 1, \cdots, n$) [28–30]. We briefly review the classification in [30].

We impose the following two constraints on the superpotentials.

1. The IR SCFT has a unique normalizable vacuum. This implies that the superpotential $W(\Phi^{(i)})$ is compact, i.e. the equations

$$\partial_{\Phi^{(1)}} W = \cdots = \partial_{\Phi^{(n)}} W = 0 \qquad (2.22)$$

   has a unique solution

$$\Phi^{(1)} = \cdots = \Phi^{(n)} = 0. \qquad (2.23)$$

2. The theory is indecomposable, which implies that the superpotential cannot be written as a sum of two terms involving different variables, i.e. for example

$$W(\Phi^{(1)}, \cdots, \Phi^{(n)}) = W_1(\Phi^{(1)}, \cdots, \Phi^{(k)}) + W_2(\Phi^{(k+1)}, \cdots, \Phi^{(n)}). \qquad (2.24)$$

Since we will focus on the IR SCFT, two different superpotentials, which define different UV theories, are regarded as IR-equivalent if the theories flow to the same IR SCFT. In particular, this implies the following two IR-equivalence relations between superpotentials.

1. If two superpotentials are related by a field redefinition compatible with quasi–homogeneity, then they are IR-equivalent.

2. If a superpotential $W$ has a variable $\Phi^{(a)}$, which appears only linearly or quadratically, then $W$ is IR-equivalent to a superpotential given by substituting equations of motion $\partial_{\Phi^{(a)}} W = 0$ into $W$.

In [30], the authors found all the possible R-charge assignments to the superfields $\Phi^{(1)}$, $\cdots$, $\Phi^{(n)}$ up to $n = 5$, which give superpotentials that satisfy the above two constraints and two equivalence relations. We will focus on the cases of $n = 1$ and 2.

For $n = 1$, the possible $R$-charges are

$$q_1 = \frac{1}{q}, \quad \text{for} \quad q \in \mathbb{Z}_{\geq 3}. \tag{2.25}$$

The superpotential is

$$W(\Phi_i) = g^{i_1 \cdots i_q} \Phi_{i_1} \cdots \Phi_{i_q}, \tag{2.26}$$

where we have suppressed the superscript. This model has been studied in [21, 22], and we refer to it as the Murugan-Stanford-Witten (MSW) model. The MSW model with a specified $q$ would be referred to as the $\mathrm{MSW}_q$ model. Some analysis of the MSW model will be reviewed in Section 2.3. For the non-disordered model ($N = 1$), this superpotential was referred as the $A_{q-1}$ superpotential in [28].

For $n = 2$, the possible R-charges are

$$\mathrm{I}_{k,l}: \quad (q_1, q_2) = \left( \frac{l-1}{kl}, \frac{1}{l} \right) \quad \text{for} \quad (k, l) \in \mathbb{Z}_{\geq 2} \times \mathbb{Z}_{\geq 3}, \tag{2.27}$$

and

$$\mathrm{II}_{k,l}: \quad (q_1, q_2) = \left( \frac{l-1}{kl-1}, \frac{k-1}{kl-1} \right) \quad \text{for} \quad (k, l) \in \mathbb{Z}_{\geq 2} \times \mathbb{Z}_{\geq 2}. \tag{2.28}$$

We will refer to them as type $\mathrm{I}_{k,l}$ and type $\mathrm{II}_{k,l}$ models. These two classes of models are overlapped, and we have the identifications

$$\mathrm{I}_{k,l} = \mathrm{II}_{k,l+\frac{1-l}{k}}. \tag{2.29}$$

Given the R-charges of the chiral superfields, we consider the most general quasi-homogeneous superpotentials up to field redefinitions. Such superpotentials would satisfy the compactness

and indecomposable conditions. If we specialize the superpotential by turning off some of the coefficients, then the superpotential might not satisfy the compactness and indecomposable conditions.

In Section 2.4, we will study and give detailed analyses of the models

$$\text{I}_{2,q}, \quad \text{I}_{3,3}, \quad \text{I}_{4,3}, \quad \text{II}_{3,4}. \tag{2.30}$$

For the non-disordered theories ($N_1 = N_2 = 1$), the type $\text{I}_{2,q}$, $\text{I}_{3,3}$, and $\text{I}_{4,3}$ superpotentials were referred as the $D_{l+1}$, $E_7$, and $J_{10}$ superpotentials, respectively, in [28]. The superpotentials of $\text{I}_{2,l}$, $\text{I}_{3,3}$ and $\text{II}_{3,4}$ do not have any exactly marginal deformations. The models $\text{I}_{3,4}$ has one-dimensional conformal manifolds. We will inspect how physical quantities (especially the chaos exponent) vary along the conformal manifolds.

## 2.3   Review of the Murugan-Stanford-Witten (MSW) model

The models with only one type of disordered chiral superfields and $A_{q-1}$ superpotential (2.26) were studied in [21,22]. Let us give a brief review following [22] of the computation of the two and four-point functions of the chiral superfields $\Phi_i$, the operator spectrum in the $\Phi_i \times \Phi_j$ OPE, and the chaos exponent of the model.

We start with the two-point function

$$\left\langle \widetilde{\Phi}^i(\widetilde{Z}_1)\Phi_j(Z_2) \right\rangle = \delta^i_j G(\langle 12 \rangle), \tag{2.31}$$

which is a function of the super-distances $\langle 12 \rangle$ and $\langle \bar{1}\bar{2} \rangle$ given in (A.9). The coordinates $Z, \widetilde{Z}$ are $Z = (y, \bar{y}, \theta, \bar{\theta})$, $\widetilde{Z} = (\tilde{y}, \bar{\tilde{y}}, \tilde{\theta}, \bar{\tilde{\theta}})$. In the leading order of the large $N$ limit, the propagators can be computed by summing over the melonic diagrams and satisfy the Schwinger-Dyson equations

$$D_3\overline{D}_3 G(\langle 13 \rangle) + qJ^2 \int d^2y_2 d^2\theta_2\, G(\langle 12 \rangle)G(\langle 32 \rangle)^{p-1} = \tilde{\theta}_{13}\bar{\tilde{\theta}}_{13}\delta(\langle 13 \rangle)\delta(\langle \bar{1}\bar{3} \rangle), \tag{2.32}$$

In the low energy (conformal) limit $E \ll J$, we can drop the first term of the equation, and solve the equations by considering the conformal ansatz

$$G(\langle 12 \rangle) = \frac{b}{|\langle 12 \rangle|^{2\Delta_\Phi}}, \tag{2.33}$$

Casting the ansatz into Dyson-Schwinger equation, one can determine the scaling dimension and the coefficient:

$$\Delta_\Phi = \frac{1}{q}, \quad b^q J^2 = \frac{1}{4\pi^2 q}. \tag{2.34}$$

In Section 2.5, we compute the same two-point function using supersymmetric localization, and find agreement with (2.33) and (2.34).

Next, we turn to the four-point function. We focus on the average four-point function which has a large $N$ expansion as

$$\frac{1}{N^2} \sum_{i,j=1}^{N} \left\langle \widetilde{\Phi}^i(\widetilde{Z}_1)\Phi_i(Z_2)\Phi_j(Z_3)\widetilde{\Phi}^j(\widetilde{Z}_4) \right\rangle = G(\langle 12 \rangle)G(\langle 43 \rangle) + \frac{1}{N}F(\widetilde{Z}_1, Z_2, Z_3, \widetilde{Z}_4), \quad (2.35)$$

where the first term is from a disconnected diagram. The leading connected four-point function $\mathcal{F}(\widetilde{Z}_1, Z_2, Z_3, \widetilde{Z}_4)$ can be computed by summing over the ladder diagrams, which gives the result

$$F(\widetilde{Z}_1, Z_2, Z_3, \widetilde{Z}_4) = \sum_{n=0}^{\infty} K^{\star n} \star F_0(\widetilde{Z}_1, Z_2, Z_3, \widetilde{Z}_4),$$

$$F_0(\widetilde{Z}_1, Z_2, Z_3, \widetilde{Z}_4) \equiv G(\langle 13 \rangle)G(\langle 42 \rangle),$$
(2.36)

where $K$ is the ladder kernel, whose action, denoted by $\star$, is given by

$$K \star F(\widetilde{Z}_1, Z_2, Z_3, \widetilde{Z}_4) \equiv \int d^2 y_a d^2 \theta_a d^2 \tilde{y}_b d^2 \tilde{\theta}_b \, \mathcal{K}(\widetilde{Z}_1, Z_2, Z_a, \widetilde{Z}_b) F(\widetilde{Z}_b, Z_a, Z_3, \widetilde{Z}_4),$$

$$\mathcal{K}(\widetilde{Z}_1, Z_2, Z_3, \widetilde{Z}_4) \equiv (p-1)J^2 G(\langle 1,3 \rangle)G(\langle 4,3 \rangle)^{q-2}G(\langle 4,2 \rangle),$$
(2.37)

and $K^{\star n}$ denotes the $n$-th power of the $\star$-product, i.e. for example $K^{\star 2} = K \star K$.

The kernel can be diagonalized by the eigenfunction

$$\mathcal{V}_{\Delta,\ell}(\widetilde{Z}_1, Z_2) = \frac{1}{|\langle 12 \rangle|^{2\Delta_\Phi - \Delta}} \left( \frac{\langle 12 \rangle}{\langle \bar{1}\bar{2} \rangle} \right)^{\frac{\ell}{2}} \quad (2.38)$$

as

$$k_{\Delta,\ell} \mathcal{V}_{\Delta,\ell}(\widetilde{Z}_1, Z_2) = \int d^2 y_a d^2 \theta_a d^2 \tilde{y}_b d^2 \tilde{\theta}_b \, \mathcal{K}(\widetilde{Z}_1, Z_2, Z_a, \widetilde{Z}_b) \mathcal{V}_{\Delta,\ell}(\widetilde{Z}_b, Z_a). \quad (2.39)$$

The eigenvalue is

$$k_{\Delta,\ell} = \frac{1 - \Delta_\Phi}{\Delta_\Phi} \frac{\Gamma(1 - \Delta_\Phi)^2 \Gamma(\frac{\ell - \Delta}{2} + \Delta_\Phi)\Gamma(\frac{\Delta + \ell}{2} + \Delta_\Phi)}{\Gamma(\Delta_\Phi)^2 \Gamma(1 + \frac{\ell - \Delta}{2} - \Delta_\Phi)\Gamma(1 + \frac{\Delta + \ell}{2} - \Delta_\Phi)}. \quad (2.40)$$

The spectrum of the operators in the $\widetilde{\Phi} \times \Phi$ OPE can be computed by solving the equation

$$k_{\Delta,\ell} = 1. \quad (2.41)$$

Each solution in the domain $\Delta \geq 1$ corresponds to a superconformal primary of dimension $\Delta$ and spin $\ell$.

Using the superconformal symmetry, we can fix the four-point function as

$$F(\widetilde{Z}_1, Z_2, Z_3, \widetilde{Z}_4) = \frac{1}{\langle 12 \rangle^{2\Delta_\Phi} \langle 43 \rangle^{2\Delta_\Phi}} \mathcal{F}(z, \bar{z}), \tag{2.42}$$

where $z$ and $\bar{z}$ are the cross ratios

$$z = \frac{\langle 12 \rangle \langle 43 \rangle}{\langle 12 \rangle \langle 42 \rangle}, \quad \bar{z} = \frac{\langle \bar{1}\bar{2} \rangle \langle \bar{4}\bar{3} \rangle}{\langle \bar{1}\bar{2} \rangle \langle \bar{4}\bar{2} \rangle}. \tag{2.43}$$

The four-point function could be expanded in the superconformal partial wave basis as

$$\mathcal{F}(z, \bar{z}) = \sum_{\ell=0}^{\infty} \int_0^\infty ds \frac{\langle \Xi_{\Delta,\ell}, \mathcal{F}_0 \rangle}{1 - k(\Delta, \ell)} \frac{\Xi_{\Delta,\ell}(z, \bar{z})}{\langle \Xi_{\Delta,\ell}, \Xi_{\Delta,\ell} \rangle}, \tag{2.44}$$

where $s = -i\Delta$, $\Xi_{\Delta,\ell}(z, \bar{z})$ is the superconformal partial wave, and $\langle \cdot, \cdot \rangle$ is the superconformal invariant inner product. We have removed the $\delta(0)$ in the inner product $\langle \Xi_{\Delta,\ell}, \Xi_{\Delta,\ell} \rangle$ in the denominator. Their explicit expressions are given in Appendix B. Using the relation between superconformal partial waves and superconformal blocks (B.2), we can rewrite the expansion as

$$\mathcal{F}(z, \bar{z}) = \sum_{\ell=0}^{\infty} \int_{-\infty}^{\infty} ds \, \rho(\Delta, \ell) \mathcal{G}_{\Delta,\ell}(z, \bar{z}), \tag{2.45}$$

where the density function $\rho(\Delta, \ell)$ is explicitly given by

$$\rho(\Delta, \ell) = \frac{\rho_{\mathrm{MFT}}(\Delta, \ell)}{1 - k(\Delta, \ell)}, \tag{2.46}$$

where $\rho_{\mathrm{MFT}}(\Delta, \ell)$ is the density function for the mean-field theory, explicitly given by

$$\begin{aligned}
\rho_{\mathrm{MFT}}(\Delta, \ell) = & \frac{\langle \Xi_{\Delta,\ell}, \mathcal{F}_0 \rangle \, \mathcal{S}_{\tilde{\Delta},\ell}}{\langle \Xi_{\Delta,\ell}, \Xi_{\Delta,\ell} \rangle} \\
= & -2^{1-2\Delta_\Phi+\ell} \csc\left(\frac{1}{2}\pi(\Delta - \ell + 2\Delta_\Phi)\right) \sin\left(\frac{1}{2}\pi(\Delta - \ell - 2\Delta_\Phi)\right) \\
& \times \frac{\Gamma(1-\Delta_\Phi)^2 \Gamma\left(\frac{1}{2}(1-\Delta+\ell)\right) \Gamma\left(\frac{1}{2}(\Delta+\ell)\right)}{\Gamma(\Delta_\Phi)^2 \Gamma\left(\frac{1}{2}(2-\Delta+\ell)\right) \Gamma\left(\frac{1}{2}(1+\Delta+\ell)\right)} \\
& \times \frac{\Gamma\left(-\frac{\Delta}{2} - \frac{\ell}{2} + \Delta_\Phi\right) \Gamma\left(\frac{1}{2}(-\Delta+\ell) + \Delta_\Phi\right)}{\Gamma\left(\frac{1}{2}(2-\Delta-\ell-2\Delta_\Phi)\right) \Gamma\left(\frac{1}{2}(2-\Delta+\ell-2\Delta_\Phi)\right)}. \tag{2.47}
\end{aligned}$$

The operator spectrum in the $\widetilde{\Phi} \times \Phi$ OPE is given by the solutions to the equation

$$k(\Delta, \ell) = 1. \tag{2.48}$$

The OPE coefficients are given by the residue of the density function. In particular, the OPE coefficient of (the bottom component of) the stress tensor multiplet $\mathcal{R}$ is given by

$$\left|c_{\widetilde{\Phi}\Phi\mathcal{R}}\right|^2 = -\frac{1}{N}\underset{\Delta=1}{\text{Res}}(\rho(\Delta,1)) = \frac{4\Delta_\Phi^2}{N(1-2\Delta_\Phi)}\,, \tag{2.49}$$

from which we compute the central charge of the IR theory

$$c = \frac{12\Delta_\Phi^2}{\left|c_{\widetilde{\Phi}\Phi\mathcal{R}}\right|^2} = N(3-6\Delta_\Phi) = \sum_{i=1}^N 6\left(\frac{1}{2}-\Delta_\Phi\right)\,. \tag{2.50}$$

We recognize that the central charge computed in this way agrees with the one obtained from the general arguments using the R-symmetry anomaly matching and the structure of $\mathcal{N} = (2,2)$ superconformal algebra [28, 39]. This central charge coincides with the central charge of $N$ copies of the $A_{q-1}$ type $\mathcal{N} = (2,2)$ minimal model, which shows up as the IR theory of the non-disordered ($N = 1$) version of the superpotential (2.26) [28]. This is because the central charge is invariant under exactly marginal deformations [40], which corresponds to deformations of the UV superpotential.

As discussed in [21], after analytic continuing of the Euclidean four-point function (2.44) to the out-of-time-order correlator in the Lorentzian signature, and taking the long time limit (chaos limit), the chaos exponent $\lambda_L$ is computed by solving the same equation (2.48) with $\Delta = 0$ and $\ell = \lambda_L$. The chaos exponent $\lambda_L$ as a function of $\Delta_\Phi$ is plotted in Figure 1. At $\Delta_\Phi = \frac{1}{3}$ ($q = 3$), the chaos exponent reaches the highest value $\lambda_L \approx 0.5824$.

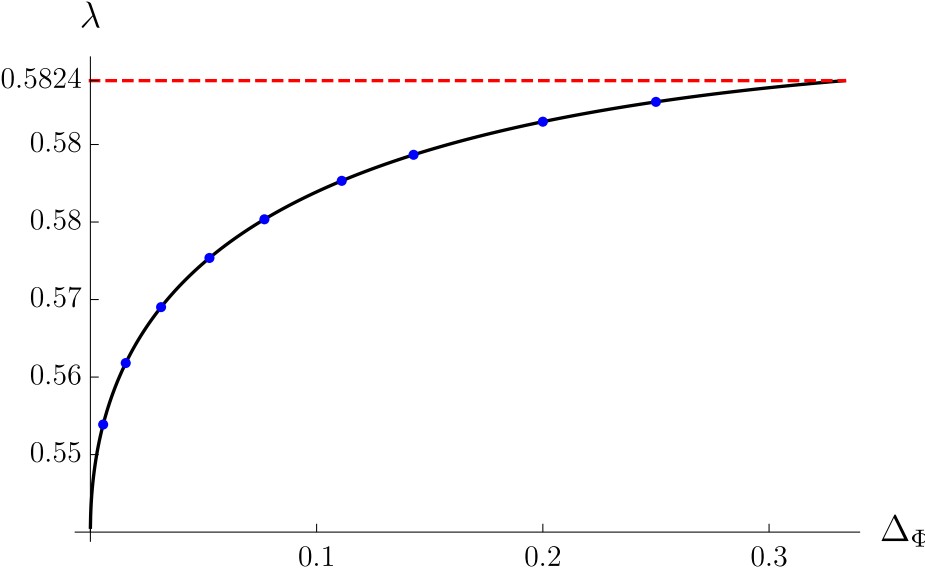

Figure 1: The chaos exponent $\lambda_L$ as function of $\Delta_\Phi = \frac{1}{q}$ in MSW model. When $\Delta_\Phi = \frac{1}{3}$, chaos exponent arrives at the maximum value 0.5824. Extrapolation is used to reach large $q$ behavior.

## 2.4 Models with two disordered chiral superfields

Let us now consider the models with two disordered chiral superfields $\Phi_i^{(1)}, \Phi_a^{(2)}$ for $i = 1, \cdots N_1$ and $a = 1, \cdots N_2$. For type $\mathrm{I}_{k,l}$ or $\mathrm{II}_{k,l}$ models, the general form of the superpotential (2.3) specializes becomes

$$W = \sum_{(m,n)\in\mathcal{I}} g^{m,n}(\Phi^{(1)})^m(\Phi^{(2)})^n \equiv \sum_{(m,n)\in\mathcal{I}} g^{i_1\cdots i_m,a_1\cdots a_n}(\Phi_{i_1}^{(1)}\cdots\Phi_{i_m}^{(1)})(\Phi_{a_1}^{(2)}\cdots\Phi_{a_n}^{(2)}), \qquad (2.51)$$

where the random coupling $g^{(i_1,\cdots,i_m),(a_1,\cdots,a_n)}$ satisfies

$$\left\langle g^{i_1\cdots i_m,a_1\cdots a_n}\bar{g}_{i_1'\cdots i_m',a_1'\cdots a_n'}\right\rangle = \frac{J_{m,n}^2}{N_1^{m+n-1}}\delta_{i_1'}^{(i_1}\cdots\delta_{i_m'}^{i_m)}\delta_{a_1'}^{(a_1}\cdots\delta_{a_n'}^{a_n)}. \qquad (2.52)$$

Note that we have changed to a different convention on the variance here comparing to (2.5). The index set $\mathcal{I}$ is given by

$$\mathcal{I} = \left\{(m,n) \in \mathbb{Z}_{\geq 0} \times \mathbb{Z}_{\geq 0} \mid mq_1 + nq_2 = 1\right\}, \qquad (2.53)$$

where $q_1$, $q_2$ are the R-charges of $\Phi^{(1)}$ and $\Phi^{(2)}$ given in (2.27) and (2.28). The large $N$ limit of these models are taken as

$$N_1, N_2 \to \infty \quad \text{fixing} \quad \lambda = \frac{N_2}{N_1}. \qquad (2.54)$$

We would follow Section 2.3, and perform the same analysis for the type $\mathrm{I}_{k,l}$ and $\mathrm{II}_{k,l}$ models as we did for the MSW model. We first consider models with general $k$, $l$, and derive general formulae for the two and four-point functions. Then we would specialize in the models in (2.30) and study the spectra and chaos exponents. To start, we consider the two-point functions

$$\left\langle\widetilde{\Phi}^{(1),i}(\widetilde{Z}_1)\Phi_j^{(1)}(Z_2)\right\rangle = \delta_j^i G_{\Phi^{(1)}}(\langle 12\rangle), \quad \left\langle\widetilde{\Phi}^{(2),a}(\widetilde{Z}_1)\Phi_b^{(2)}(Z_2)\right\rangle = \delta_b^a G_{\Phi^{(1)}}(\langle 12\rangle), \qquad (2.55)$$

where $Z = (y, \bar{y}, \theta, \bar{\theta})$ and $\widetilde{Z} = (\tilde{y}, \bar{\tilde{y}}, \tilde{\theta}, \bar{\tilde{\theta}})$, and the super-distances $\langle 12\rangle$ and $\langle \bar{1}2\rangle$ are given in (A.9). In the large $N$ limit, the two-point functions satisfy the Schwinger-Dyson equations

$$D_3\overline{D}_3 G_{\Phi^{(1)}}(\langle 13\rangle) + \int d^2y_2 d^2\theta_2\, G_{\Phi^{(1)}}(\langle 12\rangle)\Sigma_{\Phi^{(1)}}(\langle 32\rangle) = \tilde{\theta}_{13}\bar{\tilde{\theta}}_{13}\delta(\langle 13\rangle)\delta(\langle \bar{1}3\rangle),$$
$$D_3\overline{D}_3 G_{\Phi^{(2)}}(\langle 13\rangle) + \int d^2y_2 d^2\theta_2\, G_{\Phi^{(2)}}(\langle 12\rangle)\Sigma_{\Phi^{(2)}}(\langle 32\rangle) = \tilde{\theta}_{13}\bar{\tilde{\theta}}_{13}\delta(\langle 13\rangle)\delta(\langle \bar{1}3\rangle), \qquad (2.56)$$

where the self-energies $\Sigma_{\Phi^{(1)}}$ and $\Sigma_{\Phi^{(2)}}$ are

$$\Sigma_{\Phi^{(1)}}(\langle 32\rangle) = \sum_{(m,n)\in\mathcal{I}} m\lambda^n J_{m,n}^2 G_{\Phi^{(1)}}(\langle 32\rangle)^{m-1} G_{\Phi^{(2)}}(\langle 32\rangle)^n,$$
$$\Sigma_{\Phi^{(2)}}(\langle 32\rangle) = \sum_{(m,n)\in\mathcal{I}} n\lambda^{n-1} J_{m,n}^2 G_{\Phi^{(1)}}(\langle 32\rangle)^m G_{\Phi^{(2)}}(\langle 32\rangle)^{n-1}. \qquad (2.57)$$

Similar to the MSW model, in the low energy limit $E \ll J$, after ignoring the first terms of the equations in (2.56), we consider the conformal ansatz

$$G_{\Phi^{(1)}}(\langle 12 \rangle) = \frac{b_1}{|\langle 12 \rangle|^{2\Delta_1}}, \quad G_{\Phi^{(2)}}(\langle 12 \rangle) = \frac{b_2}{|\langle 12 \rangle|^{2\Delta_2}}. \tag{2.58}$$

The Schwinger-Dyson equations (2.56) fix the conformal dimensions $\Delta_1$, $\Delta_2$ by R-charges as

$$\Delta_1 = q_1, \quad \Delta_2 = q_2, \tag{2.59}$$

and impose the equations on the two-point function coefficients $b_1$, $b_2$,

$$\sum_{(m,n)\in\mathcal{I}} m\lambda^n J_{m,n}^2 b_1^m b_2^n = \frac{1}{4\pi^2}, \quad \sum_{(m,n)\in\mathcal{I}} n\lambda^{n-1} J_{m,n}^2 b_1^m b_2^n = \frac{1}{4\pi^2}. \tag{2.60}$$

The equations (2.60) admit multiple solutions. Unitarity imposes further constraints that the two-point function coefficients $b_1$ and $b_2$ are non-negative numbers,

$$b_1 \geq 0, \quad b_2 \geq 0. \tag{2.61}$$

Later in the examples, we will see that the unitarity bounds (2.61) give bounds on $\lambda$, and the model becomes non-compact when the bounds are saturated.

Next, we consider the averaged four-point functions,

$$\langle \mathcal{O}_1(\widetilde{Z}_1, Z_2)\mathcal{O}_1(\widetilde{Z}_4, Z_3)\rangle = G_{\Phi^{(1)}}(\langle 12 \rangle)G_{\Phi^{(1)}}(\langle 34 \rangle) + \frac{1}{N_1}F_{11}(\widetilde{Z}_1, Z_2, Z_3, \widetilde{Z}_4),$$

$$\langle \mathcal{O}_2(\widetilde{Z}_1, Z_2)\mathcal{O}_2(\widetilde{Z}_4, Z_3)\rangle = G_{\Phi^{(2)}}(\langle 12 \rangle)G_{\Phi^{(2)}}(\langle 34 \rangle) + \frac{1}{N_2}F_{22}(\widetilde{Z}_1, Z_2, Z_3, \widetilde{Z}_4),$$

$$\langle \mathcal{O}_1(\widetilde{Z}_1, Z_2)\mathcal{O}_2(\widetilde{Z}_4, Z_3)\rangle = \frac{1}{N_2}F_{12}(\widetilde{Z}_1, Z_2, Z_3, \widetilde{Z}_4),$$

$$\langle \mathcal{O}_2(\widetilde{Z}_1, Z_2)\mathcal{O}_1(\widetilde{Z}_4, Z_3)\rangle = \frac{1}{N_1}F_{21}(\widetilde{Z}_1, Z_2, Z_3, \widetilde{Z}_4), \tag{2.62}$$

where $\mathcal{O}_1$ and $\mathcal{O}_2$ are the bi-local operators

$$\mathcal{O}_1(\widetilde{Z}_1, Z_2) = \frac{1}{N_1}\sum_{i=1}^{N_1} \Phi^{(1),i}(\widetilde{Z}_1)\Phi_i^{(1)}(Z_2),$$

$$\mathcal{O}_2(\widetilde{Z}_1, Z_2) = \frac{1}{N_2}\sum_{a=1}^{N_2} \Phi^{(2),a}(\widetilde{Z}_1)\Phi_a^{(2)}(Z_2). \tag{2.63}$$

The four-point functions $F_{11}$, $F_{12}$, $F_{21}$, $F_{22}$ can be computed by summing over the ladder diagrams, and the result can be written in a compact form as

$$\mathcal{F}(z, \bar{z}) \equiv \begin{pmatrix} F_{11} & F_{12} \\ F_{21} & F_{22} \end{pmatrix} = \sum_{n=0}^{\infty} \begin{pmatrix} K_{11} & K_{12} \\ K_{21} & K_{22} \end{pmatrix}^{\star n} \star \begin{pmatrix} F_{11,0} & 0 \\ 0 & F_{22,0} \end{pmatrix}, \tag{2.64}$$

$F_{11,0}$, $F_{22,0}$ are the zeroth ordered disconnected ladder diagrams,

$$F_{11,0} = G_{\Phi^{(1)}}(\langle 13 \rangle) G_{\Phi^{(1)}}(\langle 42 \rangle) , \quad F_{22,0} = G_{\Phi^{(2)}}(\langle 13 \rangle) G_{\Phi^{(2)}}(\langle 42 \rangle) . \tag{2.65}$$

The matrix elements $K_{11}$, $K_{12}$, $K_{21}$, $K_{22}$ of the ladder kernel matrix are

$$
\begin{aligned}
K_{11} &= \sum_{(m,n)\in\mathcal{I}} m(m-1)\lambda^n J_{m,n}^2 G_{\Phi^{(1)}}(\langle 13 \rangle) G_{\Phi^{(1)}}(\langle 42 \rangle) G_{\Phi^{(1)}}(\langle 43 \rangle)^{m-2} G_{\Phi^{(2)}}(\langle 43 \rangle)^n , \\
K_{12} &= \sum_{(m,n)\in\mathcal{I}} mn\lambda^n J_{m,n}^2 G_{\Phi^{(1)}}(\langle 13 \rangle) G_{\Phi^{(1)}}(\langle 42 \rangle) G_{\Phi^{(1)}}(\langle 43 \rangle)^{m-1} G_{\Phi^{(2)}}(\langle 43 \rangle)^{n-1} , \\
K_{21} &= \sum_{(m,n)\in\mathcal{I}} mn\lambda^{n-1} J_{m,n}^2 G_{\Phi^{(2)}}(\langle 13 \rangle) G_{\Phi^{(2)}}(\langle 42 \rangle) G_{\Phi^{(1)}}(\langle 43 \rangle)^{m-1} G_{\Phi^{(2)}}(\langle 43 \rangle)^{n-1} , \\
K_{22} &= \sum_{(m,n)\in\mathcal{I}} n(n-1)\lambda^{n-1} J_{m,n}^2 G_{\Phi^{(2)}}(\langle 13 \rangle) G_{\Phi^{(2)}}(\langle 42 \rangle) G_{\Phi^{(1)}}(\langle 43 \rangle)^m G_{\Phi^{(2)}}(\langle 43 \rangle)^{n-2} ,
\end{aligned}
\tag{2.66}
$$

which acts on $F_{11,0}$, $F_{22,0}$ in the way as in (2.37).

Consider the eigenvector:

$$\mathcal{V}_{\Delta,\ell}^T = \left( \frac{v_1}{|\langle 43 \rangle|^{2\Delta_1 - \Delta}} \left( \frac{\langle 43 \rangle}{\langle \overline{43} \rangle} \right)^{\frac{\ell}{2}} , \frac{v_2}{|\langle 43 \rangle|^{2\Delta_2 - \Delta}} \left( \frac{\langle 43 \rangle}{\langle \overline{43} \rangle} \right)^{\frac{\ell}{2}} \right) , \tag{2.67}$$

the ladder kernel matrix acts on $\mathcal{V}_{\Delta,\ell}^T$ as a $2\times 2$ matrix,

$$
\begin{aligned}
\begin{pmatrix} K_{11} & K_{12} \\ K_{21} & K_{22} \end{pmatrix} \star \mathcal{V}_{\Delta,\ell} &= \sum_{(m,n)\in\mathcal{I}} J_{m,n}^2 \begin{pmatrix} m(m-1)\lambda^n b_1^m b_2^n k_1 & mn\lambda^n b_1^{m+1} b_2^{n-1} k_1 \\ mn\lambda^{n-1} b_1^{m-1} b_2^{n+1} k_2 & n(n-1)\lambda^{n-1} b_1^m b_2^n k_2 \end{pmatrix} \mathcal{V}_{\Delta,\ell} \\
&\equiv \begin{pmatrix} k_{11} & k_{12} \\ k_{21} & k_{22} \end{pmatrix} \mathcal{V}_{\Delta,\ell} ,
\end{aligned}
\tag{2.68}
$$

where $k_1$, $k_2$ are functions of the conformal dimension $\Delta$ and spin $\ell$,

$$k_i(\Delta, \ell) = 4\pi^2 (-1)^{\ell+1} \frac{\Gamma(1-\Delta_i)^2 \Gamma\left(\frac{\ell-\Delta}{2} + \Delta_i\right) \Gamma\left(\frac{\ell+\Delta}{2} + \Delta_i\right)}{\Gamma(\Delta_i)^2 \Gamma\left(1 + \frac{\ell-\Delta}{2} - \Delta_i\right) \Gamma\left(1 + \frac{\ell+\Delta}{2} - \Delta_i\right)} . \tag{2.69}$$

We denote the eigenvalues of this matrix by $k_+(\Delta, \ell)$ and $k_-(\Delta, \ell)$. The four-point function can be expanded in the superconformal partial waves as

$$
\begin{aligned}
\mathcal{F}(z, \bar{z}) &= \sum_{\ell=0}^{\infty} \int_0^{\infty} ds \frac{1}{(1 - k_+(\Delta, \ell))(1 - k_-(\Delta, \ell))} \frac{\Xi_{\Delta,\ell}(z, \bar{z})}{\langle \Xi_{\Delta,\ell}, \Xi_{\Delta,\ell} \rangle} \\
&\quad \times \begin{pmatrix} 1 - k_{22} & k_{12} \\ k_{21} & 1 - k_{11} \end{pmatrix} \begin{pmatrix} \langle \Xi_{\Delta,\ell}, \mathcal{F}_{11,0} \rangle & 0 \\ 0 & \langle \Xi_{\Delta,\ell}, \mathcal{F}_{22,0} \rangle \end{pmatrix} ,
\end{aligned}
\tag{2.70}
$$

where again we have removed the $\delta(0)$ in the inner product $\langle \Xi_{\Delta,\ell}, \Xi_{\Delta,\ell} \rangle$ in the denominator.

Using the shadow symmetry of the superconformal partial wave, the $s$-integral can be completed to the entire real line $\mathbb{R}$. The conformal block expansion of the four-point function is obtained by pulling the $s$-contour to the right. The operator spectrum in the $\widetilde{\Phi}^{(a)} \times \Phi^{(b)}$ OPE is given by the solution to the equation

$$(1 - k_+(\Delta, \ell))(1 - k_-(\Delta, \ell)) = 0 \,. \tag{2.71}$$

The OPE coefficients between the disordered chiral superfields $\Phi^{(1)}$, $\Phi^{(2)}$ and the bottom component of the stress tensor multiplet $\mathcal{R}$ are extracted from the residues

$$\begin{aligned}
\left| c_{\widetilde{\Phi}^{(1)}\Phi^{(1)}\mathcal{R}} \right|^2 &= -\frac{1}{N_1} \operatorname*{Res}_{\Delta=1} \left( \frac{1 - k_{22}(\Delta, \ell)}{(1 - k_+(\Delta, \ell))(1 - k_-(\Delta, \ell)} \frac{\langle \Xi_{\Delta,\ell}, \mathcal{F}_{11,0} \rangle}{\langle \Xi_{\Delta,\ell}, \Xi_{\Delta,\ell} \rangle} \bigg|_{\ell=1} \right), \\
\left| c_{\widetilde{\Phi}^{(2)}\Phi^{(2)}\mathcal{R}} \right|^2 &= -\frac{1}{N_2} \operatorname*{Res}_{\Delta=1} \left( \frac{1 - k_{11}(\Delta, \ell)}{(1 - k_+(\Delta, \ell))(1 - k_-(\Delta, \ell)} \frac{\langle \Xi_{\Delta,\ell}, \mathcal{F}_{22,0} \rangle}{\langle \Xi_{\Delta,\ell}, \Xi_{\Delta,\ell} \rangle} \bigg|_{\ell=1} \right).
\end{aligned} \tag{2.72}$$

We also obtain the central charge of the IR SCFT

$$c = \frac{12\Delta_{\Phi^{(1)}}^2}{|c_{\widetilde{\Phi}^{(1)}\Phi^{(1)}\mathcal{R}}|^2} = \frac{12\Delta_{\Phi^{(2)}}^2}{|c_{\widetilde{\Phi}^{(2)}\Phi^{(2)}\mathcal{R}}|^2} \,. \tag{2.73}$$

For the examples (2.30) that will be studied in details in the following subsubsections, we show that (2.73) simplifies to

$$c = 6N_1 \left( \frac{1}{2} - q_1 \right) + 6N_2 \left( \frac{1}{2} - q_2 \right), \tag{2.74}$$

which is consistent with the R-symmetry anomaly matching and the IR $\mathcal{N} = (2,2)$ super-conformal algebra, and is independent of the couplings (coefficients) in the superpotential as expected from the Zamolodchikov $c$-theorem [40]. Finally, similar to the MSW model, the chaos exponent $\lambda_L$ can be computed by solving the equation (2.71) with $\Delta = 0$ and $\ell = \lambda_L$. For the examples we studied below, the chaos exponents are bounded above by

$$\lambda_L \leq 0.5824 \,, \tag{2.75}$$

where the upper bound is the chaos exponent for the $\mathrm{MSW}_3$ model.

In the following subsections, we will specialize the above analysis of the two and four-point functions to the models (2.30).

### 2.4.1 $\mathrm{I}_{2,q}$ type

For $\mathrm{I}_{2,q}$ model, which is also a disordered generalization of $D_q$ type model, the superpotential is:

$$W = g^{i_1 j_2, a} \Phi_{i_1}^{(1)} \Phi_{i_2}^{(1)} \Phi_a^{(2)} + g^{a_1 \cdots a_q} \Phi_{a_1}^{(2)} \cdots \Phi_{a_q}^{(2)}, \tag{2.76}$$

which means the index set $\mathcal{I}$ is

$$\mathcal{I} = \{(2,1),(0,q)\}\,. \tag{2.77}$$

Following the discussion in Section 2.1, the field redefinitions of the bilocal superfields give the equivalent relations

$$\begin{pmatrix} J_{2,1}^2 \\ J_{0,q}^2 \end{pmatrix} \sim \begin{pmatrix} \lambda_1^2 \lambda_2 J_{2,1}^2 \\ \lambda_2^q J_{0,q}^2 \end{pmatrix}, \tag{2.78}$$

and we use it to set

$$J_{2,1} = J_{0,q} \equiv J\,, \tag{2.79}$$

where $J$ is a dimensionful overall coupling that sets the energy scale of the theory. The physical observables in the IR ($E \ll J$) SCFT are independent of $J$.

The conformal dimensions and R-charges of the chiral superfields $\Phi^{(1)}$ and $\Phi^{(2)}$ are

$$\Delta_1 = q_1 = \frac{q-1}{2q}\,, \quad \Delta_2 = q_2 = \frac{1}{q}\,, \tag{2.80}$$

Specializing the equations (2.60) for the two-point function coefficients $b_1$ and $b_2$ gives

$$2\lambda J_{2,1}^2 b_1^2 b_2 = \frac{1}{4\pi^2}\,, \quad q\lambda^{q-1} J_{0,q}^2 b_2^q + J_{2,1}^2 b_1^2 b_2 = \frac{1}{4\pi^2}\,. \tag{2.81}$$

When $\lambda < \frac{1}{2}$, all the solutions to (2.81) violate the unitarity bounds (2.61). When $\lambda \geq \frac{1}{2}$, there is a unique solution to (2.81) that satisfies the unitarity bounds (2.61):

$$b_1 = \frac{2^{\frac{3(1-q)}{2q}} q^{\frac{1}{2q}} \pi^{\frac{1-q}{q}} J_{0,q}^{\frac{1}{q}}}{J_{2,1}(2\lambda-1)^{\frac{1}{2q}}}\,, \quad b_2 = \frac{(2\lambda-1)^{\frac{1}{q}}}{2^{\frac{3}{q}} \pi^{\frac{2}{q}} q^{\frac{1}{q}} J_{0,q}^{\frac{2}{q}} \lambda}\,. \tag{2.82}$$

At $\lambda = 1/2$, the equations (2.81) imply $J_{0,q} = 0$, and the theory becomes non-compact. The formula (2.73) gives the central charge of the theory

$$\frac{c}{N_1} = \frac{3}{2} + 3\left(1 - \frac{2}{q}\right)\left(\lambda - \frac{1}{2}\right) \geq \frac{3}{2}\,. \tag{2.83}$$

The kernel of the theory is

$$\begin{pmatrix} k_{11} & k_{12} \\ k_{21} & k_{22} \end{pmatrix} = \begin{pmatrix} 2\lambda b_1^2 b_2 J_{2,1}^2 k_1(\Delta,\ell) & 2\lambda b_3^3 J_{2,1}^2 k_1(\Delta,\ell) \\ 2b_1 b_2^2 J_{2,1}^2 k_2(\Delta,\ell) & q(q-1)\lambda^{q-1} b_2^q J_{0,q}^2 k_2(\Delta,\ell)\,, \end{pmatrix} \tag{2.84}$$

where $k_1(\Delta,\ell)$ and $k_2(\Delta,\ell)$ are given in (2.69). The equation (2.71) for the operator spectrum in the OPE can be explicitly written down as

$$1 + \frac{4\pi^2(2\lambda - 2\lambda q + q - 1)k_2(\Delta,\ell) - k_1(\Delta,\ell)\left((2\lambda - 2\lambda q + q + 1)k_2(\Delta,\ell) + 8\pi^2\lambda\right)}{32\pi^4\lambda} = 0\,, \tag{2.85}$$

where we have substituted $b_1$ and $b_2$ by using the equations (2.81).

The chaos exponent $\lambda_L$ can be computed by solving the equation (2.85) with $\Delta = 0$ and $\ell = \lambda_L$. The result is shown in Figure 2.

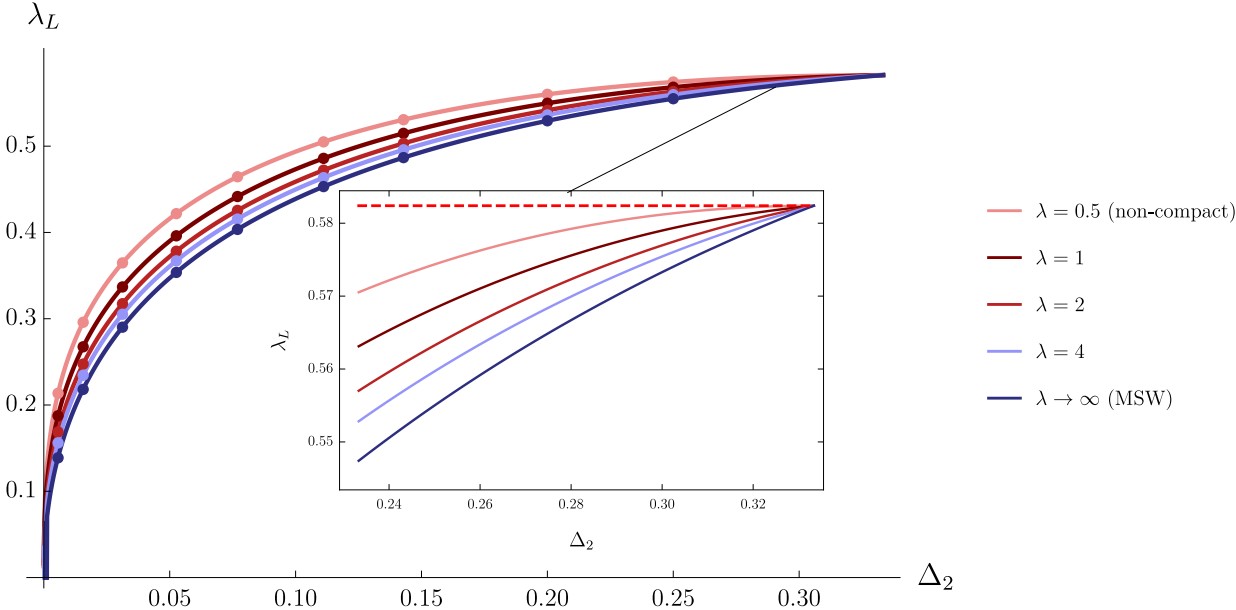

Figure 2: The chaos exponent for the $I_{2,q}\,(D_q)$ type model as a function of flavor ratio $\lambda \geq \frac{1}{2}$ and $\Delta_2 = \frac{1}{q}$. For a fixed $\lambda$, the chaos exponent grows monotonically in a similar way to the MSW model ($\lambda \to \infty$). For a fixed $\Delta_2 = \frac{1}{q}$, the chaos exponent decreases with the growth of $\lambda$. The dotted red line 0.5824 in the subfigure is the upper bound for the MSW model, which turns out to be also the upper bound for $I_{2,q}$ type model. The dots stand for the integer value of $q$, and extrapolation is used for the general value $q$.

### 2.4.2  $I_{3,3}$ type

The $I_{3,3}$ type (aka $E_7$ type) superpotential is:

$$W = g^{i_1 i_2 i_3, a} \Phi^{(1)}_{i_1} \Phi^{(1)}_{i_2} \Phi^{(1)}_{i_3} \Phi^{(2)}_a + g^{a_1 a_2 a_3} \Phi^{(2)}_{a_1} \Phi^{(2)}_{a_2} \Phi^{(2)}_{a_3}, \tag{2.86}$$

and we have the index set

$$\mathcal{I} = \{(3,1), (0,3)\}. \tag{2.87}$$

Again, the field redefinitions of the bilocal superfields give the equivalent relations

$$\begin{pmatrix} J^2_{3,1} \\ J^2_{0,3} \end{pmatrix} \sim \begin{pmatrix} \lambda_1^3 \lambda_2 J^2_{3,1} \\ \lambda_2^3 J^2_{0,3} \end{pmatrix}. \tag{2.88}$$

We use it to set the variances of the random couplings to

$$J_{3,1} = J_{0,3} \equiv J \,. \tag{2.89}$$

The conformal dimensions and R-charges of the chiral superfields $\Phi^{(1)}$ and $\Phi^{(2)}$ are

$$\Delta_1 = q_1 = \frac{2}{9}, \quad \Delta_2 = q_2 = \frac{1}{3}, \tag{2.90}$$

The equations (2.60) for the two-point function coefficients $b_1$ and $b_2$ become

$$3\lambda J_{3,1}^2 b_1^3 b_2 = \frac{1}{4\pi^2}, \quad 3\lambda^2 J_{0,3}^2 b_2^3 + J_{3,1}^2 b_1^3 b_2 = \frac{1}{4\pi^2} \,. \tag{2.91}$$

When $\lambda \geq 3$, there is a unique solution to (2.91) that satisfies the unitarity bounds (2.61):

$$b_1 = \frac{J_{0,3}^{\frac{2}{9}}}{2^{\frac{4}{9}} 3^{\frac{1}{9}} \pi^{\frac{4}{9}} J_{3,1}^{\frac{2}{3}} (3\lambda - 1)^{\frac{1}{9}}}, \quad b_2 = \frac{(3\lambda - 1)^{\frac{1}{3}}}{(6\pi)^{\frac{2}{3}} \lambda J_{0,3}^{\frac{2}{3}}} \,. \tag{2.92}$$

At $\lambda = 1/3$, the equations (2.91) imply $J_{0,3} = 0$, and the theory becomes non-compact. When $\lambda < \frac{1}{3}$, (2.91) does not admit any unitary solutions. [3] From (2.73), the central charge of the theory is

$$\frac{c}{N_1} = 2 + \left( \lambda - \frac{1}{3} \right) \geq 2 \,. \tag{2.93}$$

The kernel of the theory is

$$\begin{pmatrix} k_{11} & k_{12} \\ k_{21} & k_{22} \end{pmatrix} = \begin{pmatrix} 6b_1^3 b_2 \lambda J_{3,1}^2 k_1(\Delta, \ell) & 3b_1^4 \lambda J_{3,1}^2 k_1(\Delta, \ell) \\ 3b_1^2 b_2^2 J_{3,1}^2 k_2(\Delta, \ell) & 6b_2^3 \lambda^2 J_{0,3}^2 k_2(\Delta, \ell) \end{pmatrix} \,. \tag{2.94}$$

The equation (2.71) for the operator spectrum can be explicitly written down as

$$1 + \frac{8\pi^2(1 - 3\lambda)k_2(\Delta, \ell) + k_1(\Delta, \ell)\left((12\lambda - 7)k_2(\Delta, \ell) - 24\pi^2\lambda\right)}{48\pi^4 \lambda} = 0 \,. \tag{2.95}$$

We further take a look at the chaos exponent $\lambda_L$ by solving (2.95) with $\Delta = 0$ and $\ell = \lambda_L$. The result is shown in Figure 3.

### 2.4.3  $I_{4,3}$ type

The $I_{4,3}$ model has the superpotential

$$W = g^{i_1 \cdots i_3} \Phi_{i_1}^{(1)} \cdots \Phi_{i_3}^{(1)} + g^{i_1 i_2, a_1 a_2} \Phi_{i_1}^{(1)} \Phi_{i_2}^{(1)} \Phi_{a_1}^{(2)} \Phi_{a_2}^{(2)} + g^{i, a_1 \cdots a_4} \Phi_i^{(1)} \Phi_{a_1}^{(2)} \cdots \Phi_{a_4}^{(2)}$$
$$+ g^{a_1 \cdots a_6} \Phi_{a_1}^{(2)} \cdots \Phi_{a_6}^{(2)} \,. \tag{2.96}$$

---

[3]We thanks Micha Berkooz for discussion on this point.

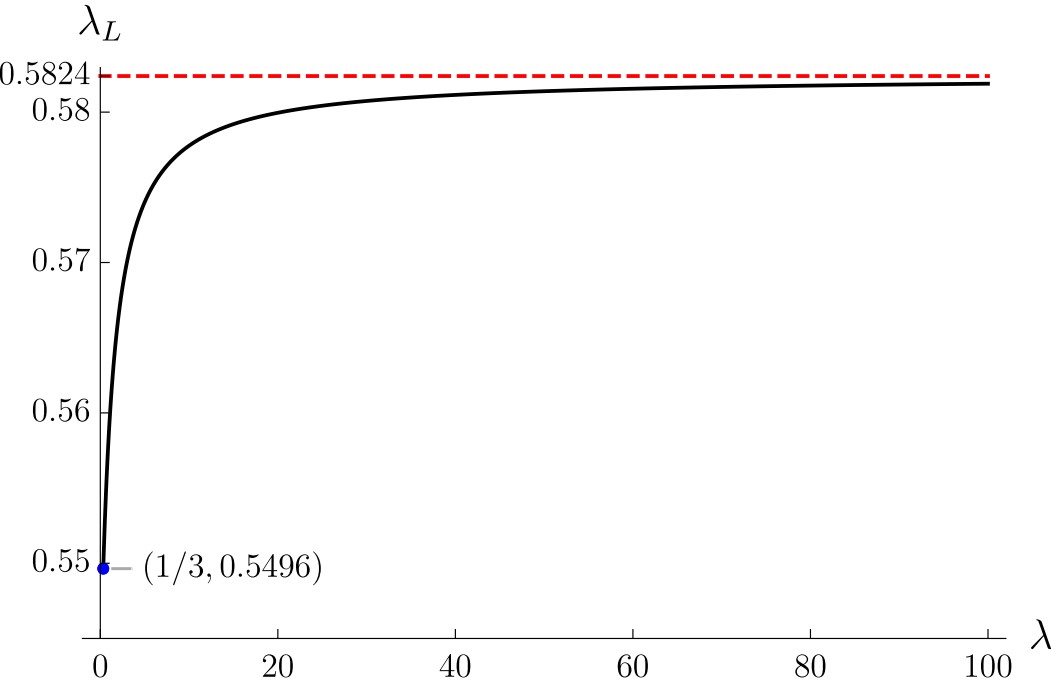

Figure 3: The chaos exponent of the $I_{3,3}$ model as function of $\lambda$. When $\lambda \to \infty$, the chaos exponent saturates 0.5824. When $\lambda \to \frac{1}{3}$, the chaos exponent equals to the non-compact lower bound 0.5496.

The index set is $\mathcal{I} = \{(3,0),\, (2,2),\, (1,4)\, (0,6)\}$. The field redefinitions of the bilocal super-fields give the equivalent relations

$$\begin{pmatrix} J_{3,0}^2 \\ J_{2,2}^2 \\ J_{1,4}^2 \\ J_{0,6}^2 \end{pmatrix} \sim \begin{pmatrix} \lambda_1^3 & 0 & 0 & 0 \\ 3a\lambda_1^2 & \lambda_1^2\lambda_2^2 & 0 & 0 \\ 3a^2\lambda_1 & 2a\lambda_1\lambda_2^2 & \lambda_1\lambda_2^4 & 0 \\ a^3 & a^2\lambda_2^2 & a\lambda_2^4 & \lambda_2^6 \end{pmatrix} \begin{pmatrix} J_{3,0}^2 \\ J_{2,2}^2 \\ J_{1,4}^2 \\ J_{0,6}^2 \end{pmatrix}. \tag{2.97}$$

We find a combination that is invariant under the above transformation

$$u^2 = \frac{3 \left( J_{2,2}^4 - 3J_{1,4}^2 J_{3,0}^2 \right)^{\frac{3}{2}}}{\left( J_{2,2}^4 - 3J_{1,4}^2 J_{3,0}^2 \right) \left[ 2J_{2,2}^6 - 9J_{1,4}^2 J_{3,0}^2 J_{2,2}^2 + 27 J_{0,6}^2 J_{3,0}^4 - 2 \left( J_{2,2}^4 - 3J_{1,4}^2 J_{3,0}^2 \right)^{\frac{3}{2}} \right]^{\frac{1}{3}}}. \tag{2.98}$$

Hence, in the IR, there is a one-dimensional conformal manifold parameterized by $u$. Equivalently, one can use the equivalence relation (2.97) to set the variances of the random couplings to

$$J_{1,4} = 0\,, \quad J_{3,0} = J_{0,6} = J\,, \quad J_{2,2} = uJ\,, \tag{2.99}$$

where $J$ is an overall dimensionful coupling. At $u = 0$, the theory factories into a tensor product of a MSW$_3$ model and a MSW$_6$ model. The parameter $u$ can be regarded as the coupling between the MSW$_3$ and the MSW$_6$ models.

Another interesting limit is $u \to \infty$. To properly take this limit, we apply the transformation (2.97) with $a = 0$, $\lambda_1 = u^{-\frac{2}{3}}$, and $\lambda_2 = u^{-\frac{1}{3}}$ on (2.99), and find

$$J_{1,4} = 0, \quad J_{3,0} = J_{0,6} = u^{-1} J, \quad J_{2,2} = J. \tag{2.100}$$

Hence, the theory becomes non-compact in the limit $u \to \infty$.

The conformal dimensions of the chiral superfields are

$$\Delta_1 = \frac{1}{3}, \quad \Delta_2 = \frac{1}{6}. \tag{2.101}$$

The two-point function coefficients $b_1$ and $b_2$ satisfy the equation

$$2\lambda^2 J_{2,2}^2 b_1^2 b_2^2 + \lambda^4 J_{1,4}^2 b_1 b_2^4 + 3 J_{3,0}^2 b_1^3 = \frac{1}{4\pi^2},$$

$$6\lambda^5 J_{0,6}^2 b_2^6 + 2\lambda J_{2,2}^2 b_1^2 b_2^2 + 4\lambda^3 J_{1,4}^2 b_1 b_2^4 = \frac{1}{4\pi^2}. \tag{2.102}$$

The equations (2.102) admit one or zero solution that satisfies the unitarity bounds (2.61) depending on the values of $\lambda$ and $u$. It is hard to determine the precise region for the existence of a unitary solution. We have tested numerically that a unitary solution exists for all values of $\lambda, u \geq 0$.

The ladder kernel is

$$\begin{pmatrix} k_{11} & k_{12} \\ k_{21} & k_{22} \end{pmatrix}$$

$$= \begin{pmatrix} 2b_1^2 \left(b_2^2 \lambda^2 J_{2,2}^2 + 3b_1 J_{3,0}^2\right) k_1(\Delta, l) & 4b_1^2 b_2 \lambda^2 \left(b_2^2 \lambda^2 J_{1,4}^2 + b_1 J_{2,2}^2\right) k_1(\Delta, l) \\ 4b_2^3 \lambda \left(b_2^2 \lambda^2 J_{1,4}^2 + b_1 J_{2,2}^2\right) k_2(\Delta, l) & 2b_2^2 \lambda \left(15 b_2^4 \lambda^4 J_{0,6}^2 + 6b_1 b_2^2 \lambda^2 J_{1,4}^2 + b_1^2 J_{2,2}^2\right) k_2(\Delta, l) \end{pmatrix}. \tag{2.103}$$

The equation (2.71) for the operator spectrum can be explicitly written down as

$$g(u) \left(2\lambda k_1(\Delta, \ell) + k_2(\Delta, \ell) \left(8 - \frac{(5\lambda + 8) k_1(\Delta, \ell)}{2\pi^2}\right)\right)$$

$$+ \frac{\left(k_1(\Delta, \ell) - 2\pi^2\right)\left(5k_2(\Delta, \ell) - 4\pi^2\right)}{8\pi^4} = 0, \tag{2.104}$$

$$g(u) = b_1^2 b_2^2 \lambda u^2 J^2,$$

where $b_1$ and $b_2$ can be solved by the equations (2.102), and $g(u)$ is a function of only the variable $u$. For general $\lambda$, $g(u)$ is a complicated function, and becomes simple when $\lambda = 1$ as

$$g(u)\big|_{\lambda=1} = \frac{1}{4\pi^2 \left(\frac{3 \times 2^{\frac{1}{3}}}{u^2} + 2\right)}. \tag{2.105}$$

The OPE spectrum depends on $u$ only through the function $g(u)$. The formula (2.73) gives the central charge of the theory:

$$\frac{c}{N_1} = 1 + 2\lambda \qquad (2.106)$$

The central charge is independent of the $g(u)$ even though the ladder kernel function is the function of the parameters. However, the chaos exponent, equivalently the Regge intercept of the theory, is the function of these parameters. When $g(u) = 0$, the two models decouple, hence one finds two roots corresponding to chaos exponent for the $\text{MSW}_3$ and the $\text{MSW}_6$, respectively.

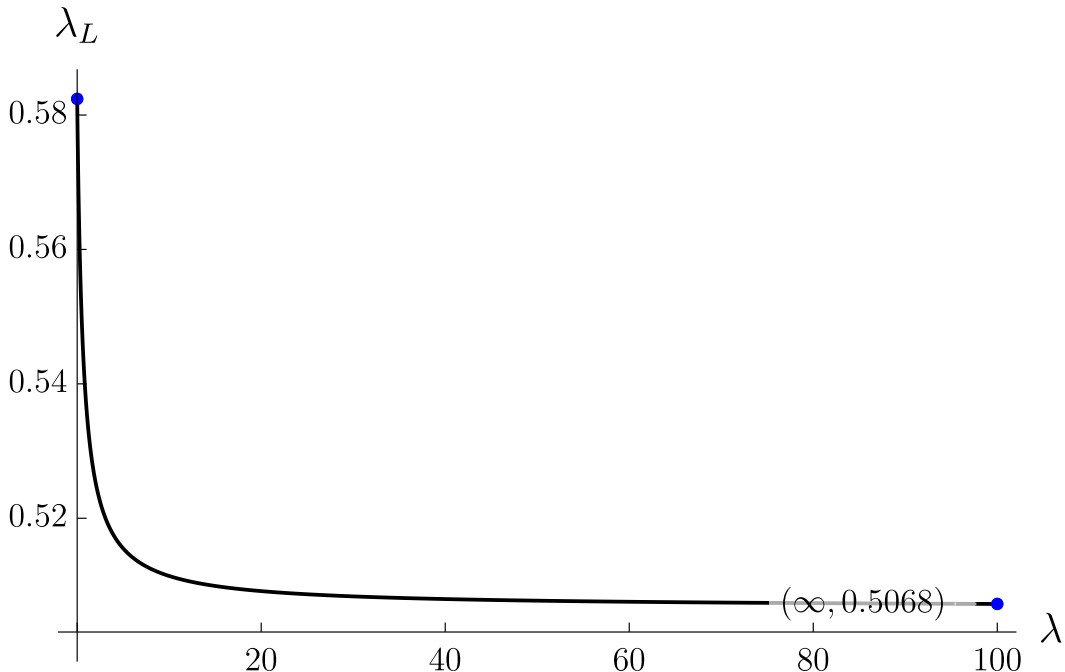

Figure 4: The chaos exponent of the $I_{4,3}$ model as function of $\lambda$ when $g(u) = 1$. When $\lambda \to 0$, the chaos exponent saturates the bound (2.75), when $\lambda \to \infty$, the chaos exponent goes to the one of the $\text{MSW}_6$.

To see the dependence between the exactly marginal deformation and chaos exponent, we first set $\lambda = 1$, then $g(u) = b_1^2 b_2^2 u^2 J^2$. One can then solve $b_1, b_2$ from the simplified equations numerically (2.102) as function of $u$. Together with (2.71), we find the relation between $u$ and $\lambda_L$, as shown in Fig(5).

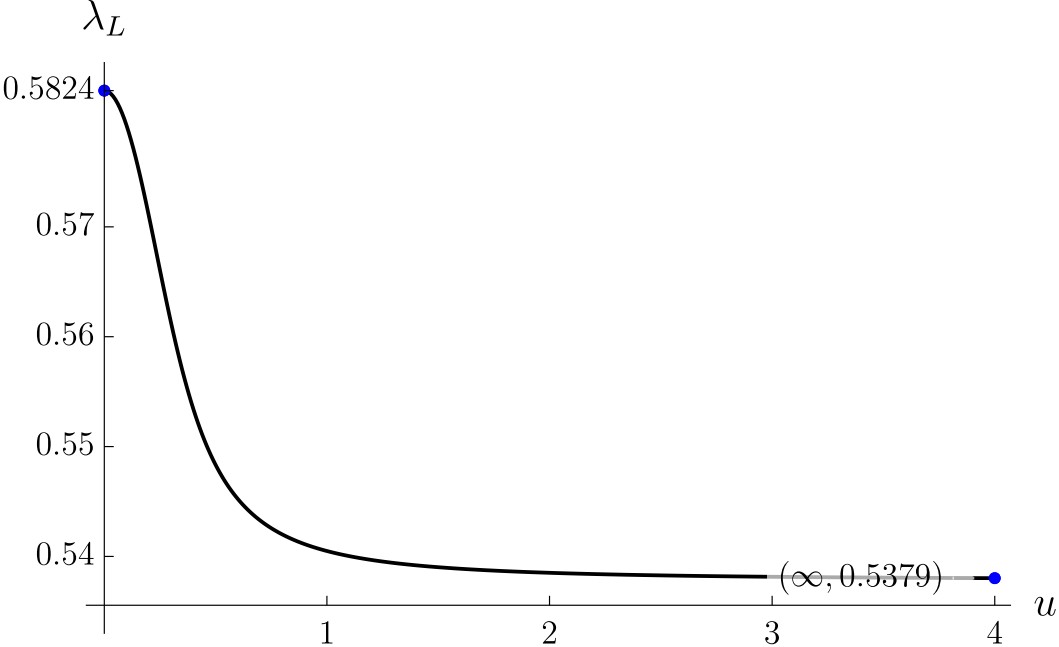

Figure 5: The chaos exponent of the $\mathrm{I}_{4,3}$ model as function of $u$ when $\lambda = 1$. When $u = 0$, the model becomes decoupled $\mathrm{MSW}_3$ and $\mathrm{MSW}_6$ with the chaos exponent equals to 0.5824. When increasing $u$, $\lambda_L$ decreases and becomes the value 0.5379 of a non-compact model when $u \to \infty$. The chaos exponent depends on $u$ weakly when $u > 2^{-\frac{1}{3}}3^{\frac{1}{2}} \approx 1.37$, where the Zamolochikov metric in (2.132) becomes negative.

### 2.4.4 $\mathrm{II}_{3,4}$ type

The $\mathrm{II}_{3,4}$ model has the superpotential:

$$W = g^{i_1\cdots i_3,a_1}\Phi^{(1)}_{i_1}\cdots\Phi^{(1)}_{i_3}\Phi^{(2)}_{a_1} + g^{i_1,a_1\cdots a_4}\Phi^{(1)}_{i_1}\Phi^{(2)}_{a_1}\cdots\Phi^{(2)}_{a_4}\,. \tag{2.107}$$

We have the index set to be

$$\mathcal{I} = \{(3,1),(1,4)\}\,, \tag{2.108}$$

and the conformal dimensions and R-charges are given by:

$$\Delta_1 = \frac{3}{11}, \quad \Delta_2 = \frac{2}{11}\,. \tag{2.109}$$

The two-point function coefficients satisfy the equations

$$3\lambda J_{3,1}^2 b_1^3 b_2 + \lambda^4 J_{1,4}^2 b_1 b_2^4 = \frac{1}{4\pi^2}, \quad J_{3,1}^2 b_1^3 b_2 + 4\lambda^3 J_{1,4}^2 b_1 b_2^4 = \frac{1}{4\pi^2}\,. \tag{2.110}$$

When $4 \geq \lambda \geq \frac{1}{3}$, the equations (2.110) admit a unique solution that satisfies the unitarity bounds (2.61). At $\lambda = \frac{1}{3}$, the equations (2.110) imply $J_{1,4} = 0$, and at $\lambda = 4$, the equations

(2.110) imply $J_{3,1} = 0$. The model is non-compact at both of these two points. When $\lambda > 4$ or $\lambda < \frac{1}{3}$, (2.110) does not have any unitary solution.

The kernel of the theory is

$$
\begin{pmatrix} k_{11} & k_{12} \\ k_{21} & k_{22} \end{pmatrix} = \begin{pmatrix} 6b_1^3 b_2 \lambda J_{3,1}^2 k_1(\Delta,\ell) & 3b_1^4 \lambda J_{3,1}^2 k_1(\Delta,\ell) + 4b_1^2 b_2^3 \lambda^4 J_{1,4}^2 k_1(\Delta,\ell) \\ 3b_1^2 b_2^2 J_{3,1}^2 k_2(\Delta,\ell) + 4b_2^5 \lambda^3 J_{1,4}^2 k_2(\Delta,\ell) & 12 b_1 b_2^4 \lambda^3 J_{1,4}^2 k_2(\Delta,\ell) \end{pmatrix}.
$$

(2.111)

From Eq.(2.73), we can obtain the central charge is

$$
\frac{c}{N_1} = \frac{3}{11}(5 + 7\lambda)
$$

(2.112)

The OPE spectrum can be explicitly written out:

$$
1 + \frac{3(-4+\lambda)}{22\pi^2} k_1(\Delta,\ell) + \frac{3-9\lambda}{11\pi^2\lambda} k_2(\Delta,\ell) + \frac{-32 + 9(8-3\lambda)\lambda}{176\pi^4\lambda} k_1(\Delta,\ell)k_2(\Delta,\ell) = 0 \quad (2.113)
$$

The chaos exponent is shown in Fig.(6)

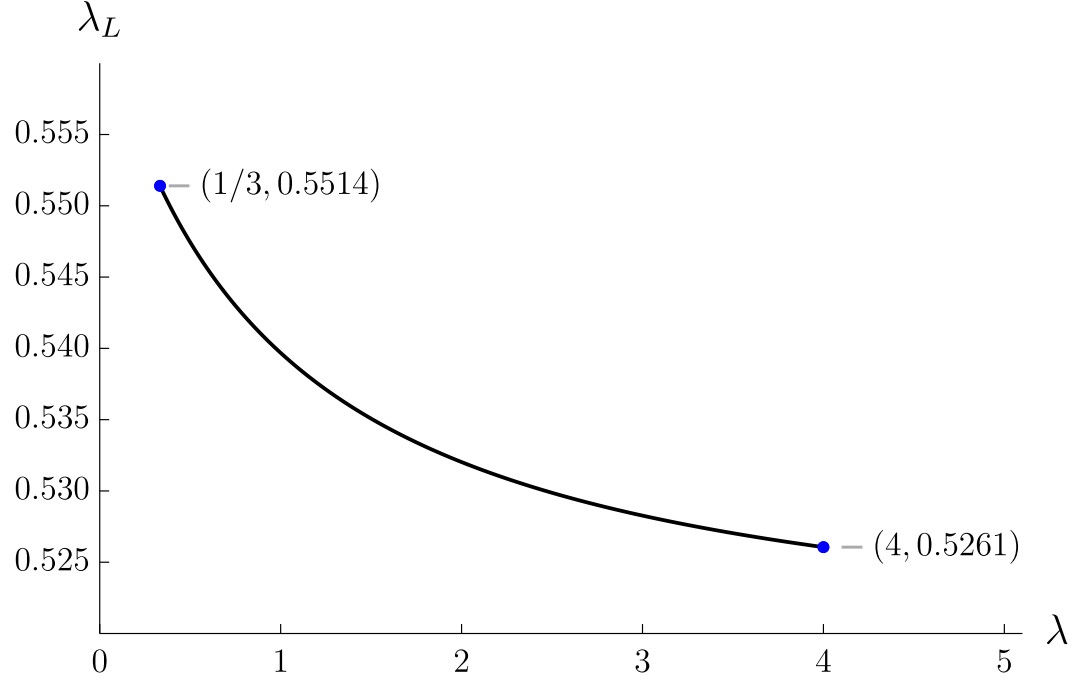

Figure 6: The chaos exponent as function of $\lambda$ in the type II$_{34}$ theory. When $\lambda = 1/3$ and 4, the model is non-compact.

## 2.5 Two-sphere partition function and two-point functions

The two-sphere partition function of the Landau-Ginzburg models can be computed by supersymmetric localization [32]. Consider a theory with $N$ chiral superfields $\Phi_i$ for $i = 1, \cdots, N$ and a superpotential $W(\Phi_i)$, the infinite-dimensional path integral localizes onto constant field configurations and becomes a finite-dimensional integral

$$Z = \int \left( \prod_i d\phi_i d\widetilde{\phi}^i \right) e^{-4\pi i r W(\phi_i) - 4\pi i r \overline{W}(\widetilde{\phi}^i)}, \tag{2.114}$$

where $r$ is the radius of the two-sphere, and $\phi_i$, $\widetilde{\phi}^i$ are the bottom components of the chiral and anti-chiral superfields $\Phi_i$, $\widetilde{\Phi}_i$, respectively. The integration contour of the integral (2.114) is defined along the half-dimensional space given by $\widetilde{\phi}^i = \phi_i^*$ inside the space $\mathbb{C}^{2N}$ of the variables $\phi_i$'s and $\widetilde{\phi}^i$'s. A common method to evaluate the integral is to decompose the contour as a sum over Lefschetz thimbles by the Picard-Lefschetz theory (see Appendix D in [34]).

This result has been generalized to extremal correlators on the two-sphere [33, 34], which is an $n$-point function of $n-1$ chiral operators inserted at arbitrary points on the two-sphere and one anti-chiral operator inserted at the south pole. For instance, the two-point function of a chiral operator $\mathcal{O}$ at the north pole and an anti-chiral operator $\widetilde{\mathcal{O}}$ at the south pole is computed by

$$\langle \widetilde{\mathcal{O}} \mathcal{O} \rangle_{\mathrm{S}^2} = \frac{1}{Z} \int \left( \prod_i d\phi_i d\widetilde{\phi}^i \right) \widetilde{\mathcal{O}} \mathcal{O} e^{-4\pi i r W(\phi_i) - 4\pi i r \overline{W}(\widetilde{\phi}^i)}. \tag{2.115}$$

When the IR theory is an SCFT, the correlation functions on $\mathrm{S}^2$ can be conformally mapped to the correlation functions on $\mathbb{R}^2$. In particular, the two-point function on the two-sphere is related to that on the plane by

$$(2r)^{2\Delta} \langle \widetilde{\mathcal{O}} \mathcal{O} \rangle_{\mathrm{S}^2} = \lim_{x \to \infty} |x|^{2\Delta} \langle \widetilde{\mathcal{O}}(x) \mathcal{O}(0) \rangle_{\mathbb{R}^2}. \tag{2.116}$$

Now, let us apply supersymmetric localization to the disordered Landau-Ginzburg models with the superpotential (2.3). The disorder-averaged two-sphere partition function is

$$Z = \frac{1}{\mathcal{N}} \int \left( \prod_{I_1, \cdots, I_n, p} dg_p^{I_1 \cdots I_n} d\bar{g}_{p, I_1 \cdots I_n} \right) e^{-\sum_p \frac{N^{p_1 + \cdots + p_n - 1}}{J_p^2} g_p^{I_1 \cdots I_n} \bar{g}_{p, I_1 \cdots I_n}} Z(g, \bar{g}),$$

$$\mathcal{N} = \int \left( \prod_{I_1, \cdots, I_n, p} dg_p^{I_1 \cdots I_n} d\bar{g}_{p, I_1 \cdots I_n} \right) e^{-\sum_p \frac{N^{p_1 + \cdots + p_n - 1}}{J_p^2} g_p^{I_1 \cdots I_n} \bar{g}_{p, I_1 \cdots I_n}}, \tag{2.117}$$

where $Z(g, \bar{g})$ is the two-sphere partition function with fixed coupling constants $g_p^{I_1 \cdots I_n}$ and

$\bar{g}_{p,I_1\cdots I_n}$. Using supersymmetric localization, $Z(g,\bar{g})$ is computed by the integral

$$Z(g,\bar{g}) = \int \left( \prod_a \prod_i d\phi_i^{(a)} d\widetilde{\phi}^{(a),i} \right) \exp\left[ -4\pi ir \sum_{p\equiv(p_1,\cdots,p_n)\in\mathcal{I}} g_p^{I_1\cdots I_n} (\phi_{I_1}^{(1)})^{p_1} \cdots (\phi_{I_n}^{(n)})^{p_n} \right.$$
$$\left. - 4\pi ir \sum_{p\equiv(p_1,\cdots,p_n)\in\mathcal{I}} \bar{g}_{p,I_1\cdots I_n} (\widetilde{\phi}^{(1),I_1})^{p_1} \cdots (\widetilde{\phi}^{(n),I_n})^{p_n} \right]. \tag{2.118}$$

Recall our notation $I_a = (i_1,\cdots,i_{p_a})$ and $(\phi_{I_a}^{(a)})^{p_a} = \phi_{i_1}^{(a)} \cdots \phi_{i_a}^{(a)}$. Performing the $g_p^{I_1\cdots I_n}$ and $\bar{g}_{p,I_1\cdots I_n}$ integrals first, we find

$$Z = \int \left( \prod_a \prod_i d\phi_i^{(a)} d\widetilde{\phi}^{(a),i} \right) \exp\left[ -16\pi^2 V(\phi_i^{(a)}, \widetilde{\phi}^{(a),i}) \right],$$
$$V(\phi_i^{(a)}, \widetilde{\phi}^{(a),i}) = \sum_{p\equiv(p_1,\cdots,p_n)\in\mathcal{I}} \frac{r^2 J_p^2}{N^{p_1+\cdots+p_n-1}} (\phi^{(1)}\widetilde{\phi}^{(1)})^{p_1} \cdots (\phi^{(n)}\widetilde{\phi}^{(n)})^{p_n}. \tag{2.119}$$

Note that since the function $V(\phi_i^{(a)}, \widetilde{\phi}^{(a),i})$ with $\widetilde{\phi}^{(a),i} = (\phi_i^{(a)})^*$ is real and bounded from below, and the integral (2.119) is much easier to compute than the integral (2.114) for non-disordered theories.[4] The integral can be further simplified as

$$Z = \int \left( \prod_a \frac{2\pi^{N_a}}{\Gamma(N_a)} R_a^{2N_a-1} dR_a \right) \exp\left[ -\sum_{p\equiv(p_1,\cdots,p_n)\in\mathcal{I}} \frac{16\pi^2 r^2 J_p^2}{N^{p_1+\cdots+p_n-1}} R_1^{2p_1} \cdots R_n^{2p_n} \right], \tag{2.120}$$

where we have used the spherical coordinates with the radius $R_a^2 = \phi^{(a)}\widetilde{\phi}^{(a)}$.

The disorder-averaged sphere two-point function is

$$\langle \widetilde{\mathcal{O}}\mathcal{O} \rangle_{S^2} = \frac{1}{\mathcal{N}Z} \int \left( \prod_{I_1,\cdots,I_n,p} dg_p^{I_1\cdots I_n} d\bar{g}_{p,I_1\cdots I_n} \right) e^{-\sum_p \frac{N^{p_1+\cdots+p_n-1}}{J_p^2} g_p^{I_1\cdots I_n}\bar{g}_{p,I_1\cdots I_n}}$$
$$\times \int \left( \prod_a \prod_{I_a} d\phi_{I_a}^{(a)} d\widetilde{\phi}^{(a),I_a} \right) \widetilde{\mathcal{O}}\mathcal{O}$$
$$\times \exp\left[ -4\pi ir \sum_{p\equiv(p_1,\cdots,p_n)\in\mathcal{I}} g_p^{I_1\cdots I_n} (\phi_{I_1}^{(1)})^{p_1} \cdots (\phi_{I_n}^{(n)})^{p_n} \right.$$
$$\left. - 4\pi ir \sum_{p\equiv(p_1,\cdots,p_n)\in\mathcal{I}} \bar{g}_{p,I_1\cdots I_n} (\widetilde{\phi}^{(1),I_1})^{p_1} \cdots (\widetilde{\phi}^{(n),I_n})^{p_n} \right]. \tag{2.121}$$

Note that (2.121) is more precisely an annealed disordered sphere two-point function, meaning that the disorder averages in the numerator and denominator are performed separately.

---

[4]We thank Sungjay Lee for a discussion on this point.

This definition allows us to compute the two-point functions exactly in the following examples.

As discussed in the previous subsection, the variances that cannot be fixed by field redefinitions parameterize the IR conformal manifold. Similar to the non-disordered model in [41,33], we can compute the Zamolodchikov metric of the IR conformal manifold by taking derivatives of the two-sphere partition function as

$$g_{p_1 p_2} = -\partial_{J_{p_1}} \partial_{J_{p_2}} \log Z \,. \tag{2.122}$$

In the following, we compute the two-sphere partition functions and the two-point functions of chiral superfields in the MSW, $I_{2,q}$, and $I_{3,3}$ model, and compute the Zamolodchikov metric of the $I_{4,3}$ model.

**Supersymmetric localization in the MSW, $I_{2,q}$, and $I_{3,3}$ models** The two-sphere partition function of the MSW model with the superpotential (2.26) is

$$
\begin{aligned}
Z_{\mathrm{MSW}_q} &= \frac{1}{\mathcal{N}} \int \Big( \prod_{i_1, \cdots, i_p} dg^{i_1 \cdots i_q} d\bar{g}_{i_1 \cdots i_q} \Big) e^{-\frac{N^{q-1}}{J^2}|g^{i_1 \cdots i_q}|^2} \int \Big( \prod_i d\phi_i d\widetilde{\phi}^i \Big) \\
&\quad \times \exp \Big( -4\pi i r g^{i_1 \cdots i_q} \phi_1 \cdots \phi_{i_q} - 4\pi i r \bar{g}_{i_1 \cdots i_q} \widetilde{\phi}^1 \cdots \widetilde{\phi}^{i_q} \Big) \\
&= \int \Big( \prod_i d\phi_i d\widetilde{\phi}^i \Big) \exp \Big( -\frac{16\pi^2 r^2 J^2}{N^{q-1}} (\phi_i \widetilde{\phi}^i)^q \Big) \\
&= \frac{16^{-\frac{N}{q}} N^{N-\frac{N}{q}} \pi^{N-\frac{2N}{q}} J^{-\frac{2N}{q}} r^{-\frac{2N}{p}} \Gamma\Big(\frac{N+q}{q}\Big)}{\Gamma(N+1)} \,.
\end{aligned}
\tag{2.123}
$$

Next, we compute the disorder-averaged sphere two-point function,

$$
\begin{aligned}
\langle \widetilde{\phi}^i \phi_j \rangle_{\mathrm{S}^2}^{\mathrm{MSW}_q} &= \frac{1}{\mathcal{N} Z_{\mathrm{MSW}_q}} \int \Big( \prod_{i_1, \cdots, i_q} dg^{i_1 \cdots i_q} d\bar{g}_{i_1 \cdots i_q} \Big) e^{-\frac{N^{q-1}}{J^2}|g^{i_1 \cdots i_q}|^2} \int \Big( \prod_i d\phi_i d\widetilde{\phi}^i \Big) \\
&\quad \times \widetilde{\phi}^i \phi_j \exp \Big( -4\pi i r g^{i_1 \cdots i_q} \phi_1 \cdots \phi_{i_q} - 4\pi i r \bar{g}_{i_1 \cdots i_q} \widetilde{\phi}^1 \cdots \widetilde{\phi}^{i_q} \Big) \\
&= \frac{\delta_j^i \Gamma\Big(\frac{N+1}{q}\Big)}{16^{\frac{1}{q}} \pi^{\frac{2}{q}} J^{\frac{2}{q}} N^{\frac{1}{q}} r^{\frac{2}{q}} \Gamma\Big(\frac{N}{q}\Big)} \,.
\end{aligned}
\tag{2.124}
$$

In the large $N$ limit, the result becomes

$$
\langle \widetilde{\phi}^i \phi_j \rangle_{\mathrm{S}^2}^{\mathrm{MSW}_q} = \frac{\delta_j^i}{(2\pi)^{\frac{2}{q}} q^{\frac{1}{q}} J^{\frac{2}{q}} (2r)^{\frac{2}{q}}} + \mathcal{O}(N^{-1}) \,.
\tag{2.125}
$$

Mapping the two-point function from $\mathrm{S}^2$ to $\mathbb{R}^2$ using (2.116), we find

$$\left\langle \widetilde{\phi}^i(x)\phi_j(0)\right\rangle^{\mathrm{MSW}_q}_{\mathbb{R}^2} = \frac{\delta^i_j}{(2\pi)^{\frac{2}{q}} p^{\frac{1}{q}} J^{\frac{2}{q}} |x|^{\frac{2}{q}}} + \mathcal{O}(N^{-1}). \tag{2.126}$$

We see that our result here nicely agrees with (2.33) and (2.34) from summing over the melonic diagrams using the Schwinger-Dyson equation.

Now, let us perform the same computation for the $\mathrm{I}_{2,q}$ and $\mathrm{I}_{3,3}$ models. For the $\mathrm{I}_{2,q}$ model, we find

$$Z_{\mathrm{I}_{2,q}} = \frac{\pi^{\frac{N_2(q-2)+N_1}{q}} N_1^{\frac{N_1(q+1)+2N_2(q-1)}{2q}} \Gamma\left(\frac{N_1}{2}\right)\Gamma\left(\frac{2N_2-N_1}{2q}\right)(rJ_{0,q})^{\frac{N_1-2N_2}{q}}}{2^{\frac{2N_1(q-1)+4N_2+q}{q}} q\Gamma(N_1)\Gamma(N_2)(rJ_{2,1})^{N_1}},$$

$$\left\langle \widetilde{\phi}^{(1),i}\phi_j^{(1)}\right\rangle^{\mathrm{I}_{2,q}}_{\mathrm{S}^2} = \delta^i_j \frac{(4\pi)^{\frac{1}{q}-1} N_1^{\frac{1}{2}\left(\frac{1}{q}-1\right)}\Gamma\left(\frac{N_1+1}{2}\right)\Gamma\left(\frac{2N_2-N_1-1}{2q}\right)(rJ_{0,q})^{\frac{1}{q}}}{\Gamma\left(\frac{N_1}{2}\right)\Gamma\left(\frac{2N_2-N_1}{2q}\right)(rJ_{2,1})}, \tag{2.127}$$

$$\left\langle \widetilde{\phi}^{(2),a}\phi_b^{(2)}\right\rangle^{\mathrm{I}_{2,q}}_{\mathrm{S}^2} = \delta^a_b \frac{N_1^{\frac{q-1}{q}}\Gamma\left(\frac{2N_2-N_1+2}{2q}\right)}{16^{\frac{1}{q}}\pi^{\frac{2}{q}} N_2\Gamma\left(\frac{2N_2-N_1}{2q}\right)(rJ_{0,q})^{\frac{2}{q}}},$$

where the two-point functions in the large $N$ limit agree with the previous result (2.82) computed by solving the Schwinger-Dyson equations.

For the $\mathrm{I}_{3,3}$ model, we find

$$Z_{\mathrm{I}_{3,3}} = \frac{\pi^{\frac{1}{9}(5N_1+3N_2)} N_1^{\frac{1}{9}(7N_1+6N_2)}\Gamma\left(\frac{N_1}{3}\right)\Gamma\left(\frac{N_2}{3}-\frac{N_1}{9}\right)(rJ_{0,3})^{\frac{2}{9}(N_1-3N_2)}}{2^{\frac{4}{9}(2N_1+3N_2)} 9\Gamma(N_1)\Gamma(N_2)(rJ_{3,1})^{\frac{2N_1}{3}}},$$

$$\left\langle \widetilde{\phi}^{(1),i}\phi_j^{(1)}\right\rangle^{\mathrm{I}_{3,3}}_{\mathrm{S}^2} = \delta^i_j \frac{\Gamma\left(\frac{N_1+1}{3}\right)\Gamma\left(\frac{3N_2-N_1-1}{9}\right)\left(\frac{rJ_{0,3}}{N_1}\right)^{\frac{2}{9}}}{2^{8/9}\pi^{4/9}\Gamma\left(\frac{N_1}{3}\right)\Gamma\left(\frac{N_2}{3}-\frac{N_1}{9}\right)(rJ_{3,1})^{\frac{2}{3}}}, \tag{2.128}$$

$$\left\langle \widetilde{\phi}^{(2),a}\phi_b^{(2)}\right\rangle^{\mathrm{I}_{3,3}}_{\mathrm{S}^2} = \delta^a_b \frac{\Gamma\left(\frac{3N_2-N_1+3}{9}\right)}{2^{\frac{4}{3}}\pi^{\frac{2}{3}} N_2\Gamma\left(\frac{N_2}{3}-\frac{N_1}{9}\right)\left(\frac{rJ_{0,3}}{N_1}\right)^{\frac{2}{3}}},$$

where the two-point functions in the large $N$ limit agree with the previous result (2.92) computed by solving the Schwinger-Dyson equations.

**Zamolodchikov metric of the $\mathrm{I}_{4,3}$ model**  Let us compute the two-sphere partition function of the $\mathrm{I}_{4,3}$ model. For simplicity, we focus on the case $N_1 = N_2 \equiv N$, and use the

parametrization of the variances (2.99). The formula (2.119) gives

$$Z_{I_{4,3}} = \int \left( \prod_i d\phi_i^{(1)} d\widetilde{\phi}^{(1),i} \prod_a d\phi_a^{(2)} d\widetilde{\phi}^{(2),a} \right) \exp \left\{ - 16\pi^2 \left[ \frac{r^2 J^2}{N^2} (\phi_i^{(1)} \widetilde{\phi}^{(1),i})^3 \right. \right.$$
$$\left. + \frac{u^2 r^2 J^2}{N^3} (\phi_i^{(1)} \widetilde{\phi}^{(1),i})^2 (\phi_a^{(2)} \widetilde{\phi}^{(2),a})^2 + \frac{r^2 J^2}{N^5} (\phi_a^{(2)} \widetilde{\phi}^{(2),a})^6 \right] \right\} \qquad (2.129)$$
$$= \frac{\pi^{2N} N^{2N}}{2\Gamma(N)^2} \int_0^\infty \int_0^\infty dR_1 dR_2 \, R_1^{N-1} R_2^{\frac{N}{2}-1} e^{-16\pi^2 N r^2 J^2 \left( R_1^3 + u^2 R_1^2 R_2 + R_2^3 \right)} ,$$

where we have changed the integration variables as $\phi_i^{(1)} \widetilde{\phi}^{(1),i} = N R_1$ and $\phi_a^{(2)} \widetilde{\phi}^{(2),a} = N\sqrt{R_2}$. The integral in the large $N$ limit can be evaluated using the saddle point approximation. The result is

$$\log Z_{I_{4,3}} = N \left[ \frac{3}{2} + \frac{1}{2} \log \left( \frac{\pi^2}{2^{\frac{13}{3}} J^2 r^2 \left( 2^{\frac{2}{3}} u^2 + 3 \right)} \right) \right] + \mathcal{O}(N^0) . \qquad (2.130)$$

For a consistency check, we take $u = 0$ of $\log Z_{I_{4,3}}$ and find that it factorizes to a sum of the log of the partition functions of the $\mathrm{MSW}_3$ and the $\mathrm{MSW}_6$ models in (2.123) in the large $N$ limit,

$$\log Z_{I_{4,3}}\big|_{u=0} = \log Z_{\mathrm{MSW}_3} + \log Z_{\mathrm{MSW}_6} . \qquad (2.131)$$

Taking $u$-derivatives, we compute the Zamolodchikov metric,

$$g_{uu} = -\frac{d^2 \log Z}{du^2} = N \frac{3 \times 2^{\frac{2}{3}} - 2 \times 2^{\frac{1}{3}} u^2}{(3 + 2^{\frac{2}{3}} u^2)^2} . \qquad (2.132)$$

Curiously, note that the metric $g_{uu}$ vanishes at $u = 2^{-\frac{1}{3}} 3^{\frac{1}{2}}$, and becomes negative when $u > 2^{-\frac{1}{3}} 3^{\frac{1}{2}}$.

Since the random couplings $g_p^{I_1 \cdots I_n}$ in the superpotential (2.3) are complex, it is tempting to replace the variance $J_p^2$ in (2.5) by $J_p \overline{J}_p$ for a complex $J_p$. This leads to the replacement of $u^2$ by $u\bar{u}$ in the two-sphere partition function (2.130). Now, the conformal manifold is complex one-dimensional, and we find the metric

$$g_{u\bar{u}} = \frac{3N}{2^{\frac{1}{3}} (3 + 2^{\frac{2}{3}} u\bar{u})^2} , \qquad (2.133)$$

which is the metric of a round two-sphere of radius $\sqrt{N/2}$. However, since $u$ always appears in the combination $u\bar{u}$, we do not know how to probe the angular direction on the conformal manifold.

We have seen that the theory becomes non-compact in the $u \to \infty$ limit. The $u = \infty$ point is at infinity on the conformal manifold with respect to the metric (2.132), but at a finite distance with respect to the metric (2.132).

# 3 Disordered gauged linear sigma models

Let us start by reviewing some basics of the gauge linear sigma models following [35] to set up our convention and notation, and along the way introduce the disordered couplings to the theory. Consider a U(1) gauge theory with chiral superfields $\Phi_i^{(1)}$ for $i = 1, \cdots, N$ of charge 1 and $\Phi_a^{(2)}$ for $a = 1, \cdots, M$ of charge $-q$. The U(1) gauge field and its superpartners form a vector superfield $V$, or equivalently a twisted chiral superfield $\Sigma = \frac{1}{\sqrt{2}} \widetilde{D}\overline{D}\mathcal{V}$. The (Euclidean) Lagrangian density of the model is

$$
\begin{aligned}
\mathcal{L} &= \mathcal{L}_{\text{kin}} + \mathcal{L}_W + \mathcal{L}_{\text{FI}} \,, \\
\mathcal{L}_{\text{kin}} &= -\int d^4\theta \left( \widetilde{\Phi}^{(1),i} e^{2\mathcal{V}} \Phi_i^{(1)} + \widetilde{\Phi}^{(2),a} e^{-2(q-1)\mathcal{V}} \Phi_a^{(2)} + \frac{1}{4e^2} \overline{\Sigma}\Sigma \right) \,, \\
\mathcal{L}_W &= -\int d^2\theta \, W(\Phi^{(1)}, \Phi^{(2)}) \Big|_{\tilde\theta = \bar{\tilde\theta} = 0} - \text{h.c.} \,, \\
\mathcal{L}_{\text{FI}} &= -\int d\theta d\bar{\tilde\theta} \, \frac{it}{2\sqrt{2}} \Sigma \big|_{\tilde\theta = \bar\theta = 0} + \int d\tilde\theta d\bar\theta \, \frac{i\bar{t}}{2\sqrt{2}} \overline{\Sigma} \big|_{\theta = \bar{\tilde\theta} = 0} \,.
\end{aligned}
\tag{3.1}
$$

The superpotential $W$ is a homogeneous polynomial given by

$$
W(\Phi^{(1)}, \Phi^{(2)}) = \Phi_a^{(2)} G^a(\Phi^{(1)}) \equiv g^{i_1 \cdots i_q, a} \Phi_{i_1}^{(1)} \cdots \Phi_{i_q}^{(1)} \Phi_a^{(2)} \,,
\tag{3.2}
$$

where the coupling constants $g_{a i_1 \cdots i_q}$ is a Gaussian random variable with mean and variance

$$
\left\langle g^{i_1 \cdots i_q, a} \right\rangle = 0 \,, \quad \left\langle g^{i_1 \cdots i_q, a} \bar{g}_{j_1 \cdots j_q, b} \right\rangle = \frac{J^2}{N^q} \delta_b^a \delta_{j_1}^{(i_1} \cdots \delta_{j_q}^{i_q)} \,.
\tag{3.3}
$$

$\mathcal{L}_{\text{FI}}$ is the Fayet-Iliopoulos term. After integrating out the Grassmann coordinates, it becomes

$$
\mathcal{L}_{\text{FI}} = rD + \frac{i\theta}{2\pi} F_{12} \,,
\tag{3.4}
$$

where $t = ir + \frac{\theta}{2\pi}$ is the Fayet-Iliopoulos parameter.

After integrating out the auxiliary fields, the potential for the bosonic fields is

$$
U = \frac{1}{2e^2} D^2 + \sum_{a=1}^{M} \left| G^a(\phi^{(1)}) \right|^2 + \sum_{i=1}^{N} \left| \sum_{a=1}^{M} \phi_a^{(2)} \frac{\partial G^a(\phi^{(1)})}{\partial \phi_i^{(1)}} \right|^2
\tag{3.5}
$$

with

$$
D = -e^2 \left( \sum_{i=1}^{N} |\phi_i^{(1)}|^2 - q \sum_{a=1}^{M} |\phi_a^{(2)}|^2 - r \right) \,,
\tag{3.6}
$$

where $\phi_i^{(1)}$ and $\phi_a^{(2)}$ denote the bottom components of the chiral superfields $\Phi_i^{(1)}$ and $\Phi_a^{(2)}$. For generic couplings $g_{i_1 \cdots i_q, a}$, the polynomials $G_a(\phi^{(1)})$ satisfy the "transverse" condition, i.e.

for any $(\phi_1^{(2)}, \cdots, \phi_M^{(2)}) \neq (0, \cdots, 0)$, the equations

$$G^a(\phi^{(1)}) = 0 = \sum_{a=1}^{M} \phi_a^{(2)} \frac{\partial G^a(\phi^{(1)})}{\partial \phi_i^{(1)}}, \tag{3.7}$$

have a common solution only for $\phi_1^{(1)} = \cdots = \phi_N^{(1)} = 0$. Note that the transverse condition is different from the compactness condition (2.22), (2.23) of the disordered Landau-Ginzburg models.

Let us analyze the low energy physics of the model. First, we assume $r > 0$. Vanishing of the D-term ($D = 0$) requires $\phi_i^{(1)}$ cannot all vanish. The transverse condition then implies $\phi_a^{(2)} = 0$. Hence, vanishing of the potential $U$ gives

$$\sum_{i=1}^{N} |\phi_i^{(1)}|^2 = r, \quad G^a(\phi^{(1)}) = 0. \tag{3.8}$$

We further divide the space of solutions of (3.8) by the U(1) gauge transformation, i.e. imposing the identification $\phi_i^{(1)} \cong \phi_i^{(1)} e^{i\theta}$. Therefore, the classical moduli space $X$ is an intersection of hypersurfaces $H_a \equiv \{G_a(\phi^{(1)}) = 0\}$ inside the complex project space $\mathbf{CP}^{N-1}$ with the projective coordinates $\phi_i^{(1)}$. After integrating out the massive fields $\phi_a^{(2)}$, the low energy effective theory is a sigma model with target space $X$.

Next, we consider the case $r < 0$. Vanishing of the D-term requires $\phi_a^{(2)}$ cannot all vanish. The transverse condition then implies $\phi_i^{(1)} = 0$. The classical moduli space is then a $\mathbf{CP}^{M-1}$ with the projective coordinates $\phi_a^{(2)}$. For $q > 2$, the massless fields are the $\phi_i^{(1)}$ and the oscillations tangent to the $\mathbf{CP}^{M-1}$. For $q = 2$, some parts of the $\phi_i^{(1)}$ become massive. The low energy effective theory is a hybrid Landau-Ginzburg/sigma model on a vector bundle over $\mathbf{CP}^{M-1}$.

We will be particularly interested in the case when the IR theory is a CFT. The $\mathcal{N} = 2$ superconformal algebra contains a U(1)$_R$ affine Lie algebra. However, in general, the axial part U(1)$_L \times$ U(1)$_R$ R-symmetry is broken quantum mechanically due to a mixed anomaly with the U(1) gauge symmetry. Vanishing of such an anomaly requires

$$\frac{M}{N} = \frac{1}{q}. \tag{3.9}$$

It is expected that the IR theory is a CFT when the condition (3.9) is met. When $r > 0$, this condition also implies that the classical moduli space $X$ is a Calabi-Yau manifold; hence, the IR CFT is a Calabi-Yau sigma model. The space of the Calabi-Yau manifold $X$ becomes the conformal manifold of the IR CFT. In particular, the complex structure moduli of $X$ is parametrized by the Gaussian random coupling constants $g_{i_1 \cdots i_q, a}$ with mean and

variance given in (3.3). The ensemble average over $g_{i_1 \cdots i_q, a}$ becomes an average of the Calabi-Yau sigma models over the part of the conformal manifold corresponding to the complex structure moduli.

The theory is solvable in the large $N$ limit:

$$N \to \infty, \quad \lambda \equiv \frac{M}{N}, \quad q, \quad t, \quad J, \quad \mu \equiv e\sqrt{N}, \tag{3.10}$$

where the last two parameters $J$, $\mu$ have classical dimension one, and the other parameters are dimensionless. We relax the condition (3.9) so that $\lambda$ and $q$ are independent parameters. We focus on the two-point functions of the chiral superfields,

$$\left\langle \widetilde{\Phi}^{(1),i}(\widetilde{Z}_1)\Phi^{(1)}_j(Z_2) \right\rangle = \delta^i_j G_{\Phi^{(1)}}(\langle 12 \rangle), \quad \left\langle \widetilde{\Phi}^{(2),a}(\widetilde{Z}_1)\Phi^{(2)}_b(Z_2) \right\rangle = \delta^a_b G_{\Phi^{(1)}}(\langle 12 \rangle). \tag{3.11}$$

They satisfy the same Schwinger-Dyson equations (2.56) as the disordered Landau-Ginzburg models. We note that, in the leading order of the large $N$ limit (3.10), the propagators of the chiral superfields do not receive corrections from the loops involving the gauge field and its superpartners. It is similar to the case of the quantum electrodynamics (QED) or the $\mathbf{CP}^{N-1}$ model in two or three dimensions, where the matter propagators also do not receive loop corrections from the gauge fields in the leading order large $N$ limit.

In the low energy limit $E \ll J$, we consider the same conformal ansatz (2.58). The Schwinger-Dyson equations (2.56) imply

$$q\Delta_1 + \Delta_2 = 1, \quad \lambda = \frac{1}{q}, \quad J^2 b_1^q b_2 = \frac{q}{4\pi^2}. \tag{3.12}$$

Note importantly that we have reproduced the condition (3.9) for the absence of $U(1)_R$ symmetry anomaly, which gives evidence for the IR conformal fixed point. This gives additional evidence that when (3.9) is satisfied the IR theory is conformal. The dimensions $\Delta_1$ and $\Delta_1$ for the chiral superfields are undetermined and constrained only by the linear equation in (3.12). This does not imply that the theory is short of determinability because the chiral superfields $\Phi^{(1)}$ and $\Phi^{(2)}$ are not gauge invariant operators. The only constraint on the scaling dimensions is that to ensure the self-energy dominates in IR, the scaling dimension of $\Phi^{(1)}$ should satisfy $\Delta_1 \in (0, \frac{1}{q})$.

The natural next step is to study the four-point function of the superfields $\Phi^{(a)}$ and $\widetilde{\Phi}^{(a)}$, and extract the OPE spectrum and the chaos exponent. However, since the $\Phi^{(a)}$ and $\widetilde{\Phi}^{(a)}$ are not gauge invariant operators, the interpretation of these quantities is subtle. We leave the analysis for future work.

# 4 Summary and discussion

In this paper, we studied $\mathcal{N} = (2, 2)$ supersymmetric field theories with random couplings in the superpotential.

1. We introduced the disordered Landau-Ginzburg models, which generalize the Murugan-Stanford-Witten model by including more families of chiral superfields. The models follow a similar classification as the non-disordered Landau-Ginzburg models. In particular, with two families of chiral superfields, the model are classified as type $I_{k,l}$ and $II_{k,l}$ with R-charges given in (2.27) and (2.28).

2. We analyzed the models $I_{2,q}$, $I_{3,3}$, $I_{4,3}$, and $II_{3,4}$. From the two and four-point functions computed exactly in the large $N$ limit, we extracted the conformal dimensions of the chiral superfields $\Delta_1$ and $\Delta_2$, the central charge $c$, and the chaos exponent $\lambda_L$. The former two agree with the expectation from the IR superconformal field theories.

3. The chaos exponent $\lambda_L$ depends on the ratio $\lambda$ of the numbers of chiral superfields in each families. For the examples we studied, we plotted $\lambda_L$ against $\lambda$ in Figures 2, 3, 4, and 6. From these data, we proposed a universal upper bound $\lambda_L \lesssim 0.5824$ for the chaos exponents in the unitary disordered Landau-Ginzburg models.

4. We computed the partition functions and two-point correlation functions of the disordered Landau-Ginzburg models on a two-sphere using supersymmetric localization. In the large $N$ limit, we showed that the results on the two-point function coefficients for the MSW, $I_{2,q}$, and $I_{3,3}$ models nicely agree with those computed by summing over melonic diagrams. We also computed the Zamolodchikov metric for the $I_{4,3}$ model.

5. We introduced the disordered gauged linear sigma models, and showed that with a positive Fayet-Iliopoulos parameter and an anomalous free $U(1)_R$ symmetry, they flow to the ensemble averages of Calabi-Yau sigma models over complex structure moduli.

It is important to extend our analysis of the disordered gauged linear sigma models to the four-point functions, from which we can extract many physical quantities such as the OPE spectrum and the chaos exponent. This would give as valuable information about the ensemble averages of Calabi-Yau sigma models. The ensemble average of Calabi-Yau sigma models over complex structure moduli was recently studied in [42] in the large volume limit. It was found that the averaged spectrum of scalar local operators exhibits the same statistical properties as the Gaussian orthogonal ensemble of random matrix theory. Although the averaging considered in [42] was with the uniform distribution as opposed to the Gaussian

distribution in our case, it is still interesting to compare their result with the OPE spectrum in our model.

Our studies on the disordered Landau-Ginzburg models can be straightforwardly generalized to higher dimensions. In three dimensions, the superpotential can be at most cubic in order for the theories to flow to nontrivial superconformal fixed points. With three or more families of chiral superfields, the disordered cubic superpotentials would have some random couplings whose variances are not fixed by field definitions, and the IR theories would exhibit nontrivial conformal manifolds. The OPE spectrum as a function of the coordinates on the conformal manifold could provide nontrivial data for testing the CFT distance conjecture in [43].

# Acknowledgements

We would like to thank Jin Chen for his collaboration during the early stage of this project. We thank Micha Berkooz, Sungjay Lee, and Mauricio Romo for helpful discussions. CC is partly supported by National Key R&D Program of China (NO. 2020YFA0713000). CC thanks Korea Institute for Advanced Study and the "Entanglement, Large N and Black Hole" workshop hosted by Asia Pacific Center for Theoretical Physics for hospitality during the progression of this work. XS is grateful for the hospitality of the Weizmann Institute of Science during the visit in 2023 spring.

# A $\quad \mathcal{N} = (2, 2)$ superspace

The $\mathcal{N} = (2, 2)$ superspace has the holomorphic and anti-holomorphic coordinates

$$(z, \theta, \tilde{\theta}) \,, \quad (\bar{z}, \bar{\theta}, \bar{\tilde{\theta}}) \,. \tag{A.1}$$

The super-derivatives are

$$
\begin{aligned}
D &= \frac{\partial}{\partial \theta} + \tilde{\theta} \frac{\partial}{\partial z} \,, \quad \overline{D} = \frac{\partial}{\partial \bar{\theta}} + \bar{\tilde{\theta}} \frac{\partial}{\partial \bar{z}} \,, \\
\widetilde{D} &= -\frac{\partial}{\partial \tilde{\theta}} - \theta \frac{\partial}{\partial z} \,, \quad \overline{\widetilde{D}} = -\frac{\partial}{\partial \bar{\tilde{\theta}}} - \bar{\theta} \frac{\partial}{\partial \bar{z}} \,.
\end{aligned}
\tag{A.2}
$$

The supercharges are realized by the differential operators

$$
\begin{aligned}
Q &= \frac{\partial}{\partial \theta} - \tilde{\theta} \frac{\partial}{\partial z} \,, \quad \overline{Q} = \frac{\partial}{\partial \bar{\theta}} - \bar{\tilde{\theta}} \frac{\partial}{\partial \bar{z}} \,, \\
\widetilde{Q} &= -\frac{\partial}{\partial \tilde{\theta}} + \theta \frac{\partial}{\partial z} \,, \quad \overline{\widetilde{Q}} = -\frac{\partial}{\partial \bar{\tilde{\theta}}} + \bar{\theta} \frac{\partial}{\partial \bar{z}} \,.
\end{aligned}
\tag{A.3}
$$

The integration measure for the superspace is defined as

$$d^2\theta \equiv d\theta d\bar{\theta}, \quad d^2\tilde{\theta} \equiv d\tilde{\theta}d\bar{\tilde{\theta}}. \tag{A.4}$$

A chiral superfield $\Phi$ satisfies the condition

$$\widetilde{D}\Phi = 0 = \overline{\widetilde{D}}\Phi, \tag{A.5}$$

and an anti-chiral superfield $\widetilde{\Phi}$ satisfies the condition

$$D\widetilde{\Phi} = 0 = \overline{D}\widetilde{\Phi}. \tag{A.6}$$

Hence, the chiral superfield $\Phi$ depends only on the coordinates

$$Z = (y, \bar{y}, \theta, \bar{\theta}), \quad y = z + \theta\tilde{\theta}, \quad \bar{y} = \bar{z} + \bar{\theta}\bar{\tilde{\theta}}, \tag{A.7}$$

and the anti-chiral superfield $\widetilde{\Phi}$ depends only on the coordinates

$$\widetilde{Z} = (\tilde{y}, \bar{\tilde{y}}, \tilde{\theta}, \bar{\tilde{\theta}}), \quad \tilde{y} = z - \theta\tilde{\theta}, \quad \bar{\tilde{y}} = \bar{z} - \bar{\theta}\bar{\tilde{\theta}}. \tag{A.8}$$

The super-distances are defined as the combinations

$$\langle 12 \rangle = \tilde{y}_1 - y_2 - 2\tilde{\theta}_1\theta_2, \quad \langle \bar{1}\bar{2} \rangle = \bar{\tilde{y}}_1 - \bar{y}_2 - 2\bar{\tilde{\theta}}_1\bar{\theta}_2, \tag{A.9}$$

which are annihilated by all the supercharges $Q_1 + Q_2, \widetilde{Q}_1 + \widetilde{Q}_2, \overline{Q}_1 + \overline{Q}_2, \overline{\widetilde{Q}}_1 + \overline{\widetilde{Q}}_2$.

# B    Two dimensional superconformal partial wave

When expanding the four point function in the basis of superconformal partial waves, one has:

$$\mathcal{F} = \sum_{\ell=0}^{\infty} \int_0^{\infty} ds \frac{\langle \Xi_{\Delta,\ell}, \mathcal{F}_0 \rangle}{1 - k(\Delta, \ell)} \frac{\Xi_{\Delta,\ell}(u, v)}{\langle \Xi_{\Delta,\ell}, \Xi_{\Delta,\ell} \rangle} \tag{B.1}$$

The principal series for the $\mathcal{N} = 2$ superconformal partial waves have conformal dimension $\Delta = is, s \in \mathbb{R}$. There are some related works about the bosonic partial waves in general $d$ dimensions, see [25, 44, 45].

One has two quantities to evaluate from the above expression: the inner product between zero rung ladder and conformal partial waves $\langle \Xi_{\Delta,\ell}, \mathcal{F}_0 \rangle$ and the norm of superconformal partial waves $\langle \Xi_{\Delta,\ell}, \Xi_{\Delta,\ell} \rangle$. Our strategy is to use known relations between superconformal blocks and bosonic blocks which enable us to deduce the relationship between the superconformal partial waves and bosonic conformal partial waves.

The superconformal partial wave is a linear superposition of superconformal blocks:

$$\Xi_{\Delta,\ell}(z,\bar{z}) = \mathcal{S}_{\widetilde{\Delta},\ell}\mathcal{G}_{\Delta,\ell}(z,\bar{z}) + \mathcal{S}_{\Delta,\ell}\mathcal{G}_{\widetilde{\Delta},\ell}(z,\bar{z}) \tag{B.2}$$

$\mathcal{S}_{\Delta,\ell}$ are some coefficients determined by $\Delta$ and $\ell$, and $\tilde{\Delta} = -\Delta$ is the 2 dimensional supershadow of $\Delta$. By using the shadow symmetry, we can unfold the integral:

$$\mathcal{F} = \sum_{\ell=0}^{\infty} \int_{-\infty}^{\infty} ds \frac{\langle \Xi_{\Delta,\ell}, \mathcal{F}_0 \rangle}{1 - k(\Delta,\ell)} \frac{\mathcal{S}_{\tilde{\Delta},\ell}\mathcal{G}_{\Delta,\ell}(z,\bar{z})}{\langle \Xi_{\Delta,\ell}, \Xi_{\Delta,\ell} \rangle} = \sum_{\ell=0}^{\infty} \int_{-\infty}^{\infty} ds \frac{\rho_{\mathrm{MFT}}\mathcal{G}_{\Delta,\ell}(z,\bar{z})}{1 - k(\Delta,\ell)} \tag{B.3}$$

where we have defined:

$$\rho_{\mathrm{MFT}} \equiv \frac{\langle \Xi_{\Delta,\ell}, \mathcal{F}_0 \rangle \mathcal{S}_{\tilde{\Delta},\ell}}{\langle \Xi_{\Delta,\ell}, \Xi_{\Delta,\ell} \rangle} \tag{B.4}$$

as the mean field spectral function. When $z, \bar{z} \to 0$, the superconformal block with 4 identical operator relates bosonic conformal block by

$$\mathcal{G}_{\Delta,\ell}(z,\bar{z}) = \frac{1}{|z|} G_{\Delta+1,\ell}^{1,-1}(z,\bar{z}) \tag{B.5}$$

$G_{\Delta,\ell}^{\Delta_{12},\Delta_{34}}$ is the bosonic conformal block in the four point function with primaries $\Delta_a, a = 1, \cdots, 4$, and $\Delta_{12} = \Delta_1 - \Delta_2, \Delta_{34} = \Delta_3 - \Delta_4$. The relation enables us to write superconformal partial waves in terms with bosonic conformal blocks:

$$\Xi_{\Delta,\ell}(z,\bar{z}) = \frac{1}{|z|} \left( \mathcal{S}_{\tilde{\Delta},\ell} G_{\Delta+1,\ell}^{1,-1}(z,\bar{z}) + \mathcal{S}_{\Delta,\ell} G_{1-\Delta,\ell}^{1,-1}(z,\bar{z}) \right) \tag{B.6}$$

on the other hand, since the linear combination of blocks appears in the superconformal partial wave, we must have

$$\Xi_{\Delta,\ell}(z,\bar{z}) = \frac{1}{|z|} \mathcal{N}(\Delta,\ell) \Psi_{\Delta+1,\ell}^{1,-1}(z,\bar{z}) \tag{B.7}$$

here $\Psi_{\Delta,\ell}^{\Delta_{12},\Delta_{34}}$ is conformal partial waves in the four point function with primaries $\Delta_a, a = 1, \cdots, 4$, which can be expressed as the linear combination as the bosonic conformal block:

$$\Psi_{\Delta,\ell}^{\Delta_{12},\Delta_{34}} = S_{\widehat{\Delta},\ell}^{\Delta_{34}} G_{\Delta,\ell}^{\Delta_{12},\Delta_{34}} + S_{\Delta,\ell}^{\Delta_{12}} G_{\widehat{\Delta},\ell}^{\Delta_{12},\Delta_{34}} \tag{B.8}$$

$\hat{\Delta} = 2 - \Delta$ is the bosonic shadow of $\Delta$. $\mathcal{N}(\Delta,\ell)$ is the normalization coefficient, which relates the $S_{\Delta,\ell}$ and $\mathcal{S}_{\Delta,\ell}$ by:

$$\mathcal{S}_{-\Delta,\ell} = \mathcal{N}(\Delta,\ell) S_{1-\Delta,\ell}^{\Delta_{34}=-1} \tag{B.9}$$

The shadow coefficient is given by:

$$S^{\Delta_{34}}_{\Delta,\ell} = \frac{\pi\Gamma(\Delta+\ell-1)\Gamma\left(\frac{\widehat{\Delta}+\Delta_{34}+\ell}{2}\right)\Gamma\left(\frac{\widehat{\Delta}-\Delta_{34}+\ell}{2}\right)}{\Gamma(\widehat{\Delta}+\ell)\Gamma\left(\frac{\Delta+\Delta_{34}+\ell}{2}\right)\Gamma\left(\frac{\Delta-\Delta_{34}+\ell}{2}\right)}.$$

(B.10)

the normalization of the superconformal partial waves follows from the bosonic case:

$$\langle \Xi_{\Delta,\ell}, \Xi_{\Delta',\ell'}\rangle = \mathcal{N}(\Delta,\ell)\mathcal{N}(\Delta',\ell')\left\langle \frac{1}{|z|}\Psi^{1,-1}_{\Delta+1,\ell}, \frac{1}{|z|}\Psi^{1,-1}_{\Delta'+1,\ell'}\right\rangle_{\text{SUSY}}$$

$$= \mathcal{N}(\Delta,\ell)\mathcal{N}(\Delta',\ell')\left\langle \Psi^{1,-1}_{\Delta+1,\ell}, \Psi^{1,-1}_{\Delta'+1,\ell'}\right\rangle^{1,-1}_{\text{Bosonic}}$$

$$= \mathcal{N}(\Delta,\ell)^2 n_{\Delta+1,\ell}2\pi\delta\left(s-s'\right)\delta_{\ell\ell'},$$

(B.11)

here we denote $\langle,\rangle_{\text{SUSY}}$ and $\langle,\rangle_{\text{Bosonic}}$ as SUSY/bosonic invariant inner product under properly gauge fixing, for the detail of measure after gauge fixing, refer to [25]:

$$\langle F, G\rangle_{\text{SUSY}} = \int d^2z \frac{1}{|z|^2|1-z|^2}FG = \int d^2z \frac{1}{|z|^4|1-z|^2}\left(|z|F\right)\left(|z|G\right) = \langle|z|F, |z|G\rangle^{\Delta_{12}=1,\Delta_{34}=-1}_{\text{Bosonic}}$$

(B.12)

$n_{\Delta,\ell}$ is the normalization coefficients in 2d bosonic conformal partial wave:

$$n_{\Delta,\ell} = \frac{\text{vol}\left(S^{d-2}\right)}{\text{vol}(\text{SO}(d-1))}\frac{4(2\ell+d-2)\pi\Gamma(\ell+1)\Gamma(\ell+d-2)}{2^{d-2}\Gamma\left(\ell+\frac{d}{2}\right)^2}\frac{1}{2^{2\ell}}\frac{\pi^d\Gamma(\Delta-\frac{d}{2})\Gamma(\hat{\Delta}-\frac{d}{2})}{(\Delta+\ell-1)(\hat{\Delta}+\ell-1)\Gamma(\Delta-1)\Gamma(\hat{\Delta}-1)}$$

(B.13)

We need not care about the expression of $\mathcal{N}(\Delta,\ell)$, since it cancels in the calculation of $\rho_{\text{MFT}}$ in the following context. Let us now consider the superconformal zero rung laddder diagram. For simplicity, we fix the gauge to be $\langle\widetilde{\Phi}(0)\Phi(z)\widetilde{\Phi}(1)\Phi(\infty)\rangle$. Under this gauge, the zero rung ladder becomes $|z|^{2\Delta_\Phi}$. The inner product is given as:

$$\langle \Xi_{\Delta,\ell}, \mathcal{F}_0\rangle = \mathcal{N}(\Delta,\ell)\left\langle\frac{1}{|z|}\Psi^{1,-1}_{\Delta+1,\ell}, |z|^{2\Delta_\Phi}\right\rangle_{\text{SUSY}} = \mathcal{N}(\Delta,\ell)\langle\Psi^{1,-1}_{\Delta+1,\ell}, |z|^{2\Delta_\Phi+1}\rangle^{1,-1}_{\text{Bosonic}}$$

(B.14)

Within the new gauge set up $\mathfrak{G} = \{x_1 = 0, x_2 = 1, x_5 = \infty\}$, one can evaluate the above inner product as:

$$\langle \Xi_{\Delta,\ell}, \mathcal{F}_0\rangle = \int \frac{d^2x_1\cdots d^2x_5}{\text{vol}(\text{SO}(2,2))}\frac{|x_{12}|^{\Delta_1+\Delta_2}|x_{34}|^{\Delta_3+\Delta_4}}{|x_{12}|^4|x_{34}|^4}\left(\frac{|x_{14}|}{|x_{23}|}\right)^{\Delta_{21}}\left(\frac{|x_{14}|}{|x_{13}|}\right)^{\Delta_{34}}\Psi^{\Delta_{12},\Delta_{34}}_{\Delta+1,\ell}(x_1,\ldots,x_5)F_0^{1+\frac{1}{2\Delta_\Phi}}\Bigg|_{\mathfrak{G}}$$

(B.15)

in our definition,

$$\Psi^{\Delta_{12},\Delta_{34}}_{\Delta,\ell} = \int d^dx_5 \frac{|x_{12}|^{\Delta-\Delta_1-\Delta_2}}{|x_{25}|^{\Delta_2+\Delta-\Delta_1}|x_{15}|^{\Delta_1+\Delta-\Delta_2}}\frac{|x_{34}|^{\widehat{\Delta}-\Delta_3-\Delta_4}}{|x_{35}|^{\Delta_3+\widehat{\Delta}-\Delta_4}|x_{45}|^{\Delta_4+\widehat{\Delta}-\Delta_3}}\widehat{C}_\ell(\eta)$$

(B.16)

where

$$|n|^J |m|^J \widehat{C}_J \left( \frac{n \cdot m}{|n||m|} \right) = (n^{\mu_1} \cdots n^{\mu_J} - \text{ traces }) (m_{\mu_1} \cdots m_{\mu_J} - \text{ traces })$$ 

(B.17)

and

$$\eta = \frac{|x_{15}|\,|x_{25}|}{|x_{12}|} \frac{|x_{35}|\,|x_{45}|}{|x_{34}|} \left( \frac{\vec{x}_{15}}{x_{15}^2} - \frac{\vec{x}_{25}}{x_{25}^2} \right) \cdot \left( \frac{\vec{x}_{35}}{x_{35}^2} - \frac{\vec{x}_{45}}{x_{45}^2} \right) \Bigg|_{\mathfrak{G}} = \frac{\vec{1} \cdot \vec{x}_{34}}{|x_{34}|}$$

(B.18)

And the zero rung ladder under the gauge is normalized as:

$$F_0 = \frac{|x_{12}|^{2\Delta_\Phi} |x_{34}|^{2\Delta_\Phi}}{|x_{13}|^{2\Delta_\Phi} |x_{24}|^{2\Delta_\Phi}}$$

(B.19)

Inserting Eq.(B.16)(B.17)(B.18)(B.19) into Eq.(B.15), we have:

$$\langle \Xi_{\Delta,\ell}, \mathcal{F}_0 \rangle = \mathcal{N}(\Delta, \ell) \frac{2}{\pi} \int d^2 x_3 d^2 x_4 \frac{|x_{34}|^{2\Delta_\Phi - \Delta - 2}}{|x_4|^2 |x_3|^{2\Delta_\Phi} |1 - x_4|^{2\Delta_\Phi}} (-1)^\ell \widehat{C}_\ell \left( \frac{\vec{1} \cdot \vec{x}_{34}}{|x_{34}|} \right) = \mathcal{N}(\Delta, \ell) \mathcal{I}(\Delta, \ell)$$

(B.20)

$\frac{2}{\pi}$ comes from the Berzinian under this gauge. The integral can be evaluated as:

$$\frac{2}{\pi} \int d^2 x_3 d^2 x_4 \frac{|x_{34}|^{2\Delta_\Phi - \Delta - 2}}{|x_4|^2 |x_3|^{2\Delta_\Phi} |1 - x_4|^{2\Delta_\Phi}} (-1)^\ell \widehat{C}_\ell \left( \frac{\vec{1} \cdot \vec{x}_{34}}{|x_{34}|} \right)$$

$$= \frac{2}{\pi} \int d^2 x_{43} d^2 x_4 \frac{|x_{43}|^{2\Delta_\Phi - \Delta - 2 - \ell} (x_{43}^{\mu_1} \dots x_{43}^{\mu_\ell} - \text{traces})}{|x_4|^2 |x_4 - x_{43}|^{2\Delta_\Phi} |1 - x_4|^{2\Delta_\Phi}} (e_{\mu_1} \cdots e_{\mu_J} - \text{ traces })$$

$$= \frac{2}{\pi} F(\Delta_\Phi - 1 - \frac{\Delta}{2} - \frac{\ell}{2}, -\Delta_\Phi, \ell) (e_{\mu_1} \cdots e_{\mu_\ell} - \text{ traces }) \int d^2 x_4 \frac{x_4^{\mu_1} \dots x_4^{\mu_\ell} - \text{traces}}{|x_4|^{\Delta + \ell + 2} |1 - x_4|^{2\Delta_\Phi}}$$

$$= \frac{2}{\pi} F(\Delta_\Phi - 1 - \frac{\Delta}{2} - \frac{\ell}{2}, -\Delta_\Phi, \ell) F(-\frac{\Delta}{2} - \frac{\ell}{2} - 1, -\Delta_\Phi, \ell) \widehat{C}_\ell(\vec{1})$$

(B.21)

where

$$F(a, b, s) \equiv \pi \frac{\sin \pi(a + s)}{\sin \pi(a + b + s + 1)} \frac{\Gamma(a + 1)\Gamma(b + 1)\Gamma(a + s + 1)}{\Gamma(a + b + 2)\Gamma(a + b + s + 2)\Gamma(-b)}.$$

(B.22)

$$\widehat{C}_\ell(1) = (e^{\mu_1} \cdots e^{\mu_\ell} - \text{ traces }) (e_{\mu_1} \cdots e_{\mu_\ell} - \text{ traces }) = \frac{1}{2^\ell}.$$

(B.23)

Since we have collected all the necessary data, we can get the spectral coefficient function:

$$
\begin{aligned}
\rho_{\mathrm{MFT}} &= \frac{\langle \Xi_{\Delta,\ell}, \mathcal{F}_0 \rangle \mathcal{S}_{\tilde{\Delta},\ell}}{\langle \Xi_{\Delta,\ell}, \Xi_{\Delta,\ell} \rangle} = \frac{\mathcal{N}(\Delta,\ell)^2 \mathcal{I}(\Delta,\ell) S_{1-\Delta,\ell}^{\Delta_{34}=-1}}{\mathcal{N}(\Delta,\ell)^2 n_{\Delta+1,\ell}} = \frac{\mathcal{I}(\Delta,\ell) S_{1-\Delta,\ell}^{\Delta_{34}=-1}}{n_{\Delta+1,\ell}} \\
&= -2^{1-2\Delta_\Phi+\ell} \csc\left(\frac{1}{2}\pi(\Delta-\ell+2\Delta_\Phi)\right) \sin\left(\frac{1}{2}\pi(\Delta-\ell-2\Delta_\Phi)\right) \\
&\quad \times \frac{\Gamma(1-\Delta_\Phi)^2 \Gamma\left(\frac{1}{2}(1-\Delta+\ell)\right) \Gamma\left(\frac{1}{2}(\Delta+\ell)\right)}{\Gamma(\Delta_\Phi)^2 \Gamma\left(\frac{1}{2}(2-\Delta+\ell)\right) \Gamma\left(\frac{1}{2}(1+\Delta+\ell)\right)} \\
&\quad \times \frac{\Gamma\left(-\frac{\Delta}{2}-\frac{\ell}{2}+\Delta_\Phi\right) \Gamma\left(\frac{1}{2}(-\Delta+\ell)+\Delta_\Phi\right)}{\Gamma\left(\frac{1}{2}(2-\Delta-\ell-2\Delta_\Phi)\right) \Gamma\left(\frac{1}{2}(2-\Delta+\ell-2\Delta_\Phi)\right)}
\end{aligned}
\tag{B.24}
$$

# C  Two dimensional central charge

Under the gauge $\{\bar{\theta}_1 = \theta_2 = \theta_3 = \bar{\theta}_4 = 0\}$, the superconformal block expansion of four point function for identical complex scalar reads:

$$
\mathcal{W}\left(\bar{X}_1, X_2, X_3, \bar{X}_4\right)\big|_{\bar{\theta}_1=\theta_2=\theta_3=\bar{\theta}_4=0} = \frac{\langle \bar{\phi}(x_1) \phi(x_2) \phi(x_3) \bar{\phi}(x_4) \rangle}{\langle \bar{\phi}(x_1) \phi(x_2) \rangle \langle \phi(x_3) \bar{\phi}(x_4) \rangle} = \sum_{\mathcal{O} \in \Phi \times \bar{\Phi}} |c_{\Phi\bar{\Phi}\mathcal{O}}|^2 \, \mathcal{G}_{\Delta,\ell}(u,v)
\tag{C.1}
$$

The superconformal block can be regarded as linear superposition of conformal block:

$$
\mathcal{G}_{\Delta,\ell} = G_{\Delta,\ell} + a_1(\Delta,\ell) G_{\Delta+1,\ell+1} + a_2(\Delta,\ell) G_{\Delta+1,\ell-1} + a_3(\Delta,\ell) G_{\Delta+2,\ell}
\tag{C.2}
$$

$$
\begin{aligned}
a_1 &= \frac{(\Delta+\ell)}{2(\Delta+\ell+1)} \\
a_2 &= \frac{(\Delta-\ell)}{8(\Delta-\ell+1)} \\
a_3 &= \frac{(\Delta+\ell)(\Delta-\ell)}{16(\Delta+\ell+1)(\Delta-\ell+1)}
\end{aligned}
\tag{C.3}
$$

notice that when $(\Delta,\ell) = (1,1)$, $a_2 = a_3 = 0$. In the limit of $u \to 0, v \to 1$, the conformal block goes to:

$$
G_{\Delta,\ell}(u,v) \to \frac{(-1)^\ell}{2^\ell} u^{\frac{\Delta-\ell}{2}} (1-v)^\ell
\tag{C.4}
$$

By using the OPE, we have the contribution from stress tensor in the above four point function:

$$
1 + \frac{C_{\phi\bar{\phi}T}^2}{c} \frac{V_{S^1}^2}{u^{-1}v} \left(\frac{(u+v-1)^2}{4uv} - \frac{1}{2}\right) \subset \frac{\langle \bar{\phi}(x_1) \phi(x_2) \phi(x_3) \bar{\phi}(x_4) \rangle}{\langle \bar{\phi}(x_1) \phi(x_2) \rangle \langle \phi(x_3) \bar{\phi}(x_4) \rangle}
\tag{C.5}
$$

compare with the linear combination of superconformal blocks, we have:

$$a_1(\Delta, \ell) \left| C_{\widetilde{\Phi}\Phi\mathcal{R}} \right|^2 = \frac{\left| C_{\bar{\phi}\phi T} \right|^2}{c} V_{S^1}^2 \tag{C.6}$$

where the Ward identity fixes the OPE coefficient $|C_{\bar{\phi}\phi T}|$:

$$|C_{\bar{\phi}\phi T}| = \frac{\Delta_\Phi}{\pi} \tag{C.7}$$

together with Eq.(C.3) and Eq.(C.6), we have:

$$c = \frac{12\Delta_\Phi^2}{|C_{\widetilde{\Phi}\Phi\mathcal{R}}|^2} \tag{C.8}$$

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
