# Peer review of "Disordered $\mathcal{N} = (2, 2)$ Supersymmetric Field Theories"

_SciPost Physics_

## Round 1 · Referee Report · Anonymous (Referee 1) · 2024-3-9

Strengths

  1. The paper is clear and well written.
  2. The authors study disorder-averaged supersymmetric field theories in two dimensions, giving an interesting generalisation of the analysis of SYK models. The main focus of the paper is on disordered Landau-Ginzburg models. The authors compute the central charge, the operator spectrum and the chaos exponent of these models. They use both diagrammatic and exact methods, such as localisation, with the results agreeing.
  3. The authors point out that these models admit a conformal manifold parametrised by the variances of the random couplings which lead to the disorder average, and they show that the above quantities depend non-trivially on these marginal couplings.
  4. The authors give a preliminary study of disordered GLSMs, which can be interpreted at low energies as ensemble averages of Calabi-Yau sigma models, where one averages over the complex structure moduli space.

Weaknesses

  1. The disorder average for the GLSMs uses marginal couplings which are Gaussian random variables. A more natural average might come from taking the distribution implied by the canonical metric on complex structure moduli space.
  2. The treatment of GLSMs is somewhat cursory, though the authors indicate that they will return to this question.

Report

This article aims to extend the techniques of disorder-averaging to a large class of (2,2) field theories in two dimensions. A particular focus is on theories that are disordered versions of Landau–Ginzburg models. Using large-N techniques and exact localisation results, the authors study the disorder-average two- and four-point functions of chiral fields. These give access to the central charge, the spectrum of operators, and the chaos exponent for these disordered theories.

Since these theories can have multiple chiral fields, the random couplings that appear in their superpotentials are parametrised by multiple variances. Not all of these variances can be shifted away by field redefinitions, leaving a conformal manifold of theories. By varying these variances, the authors are able to move around the conformal manifold and thus study the chaos exponent as a function of these marginal couplings.

The authors then extend their approach to gauged linear sigma models. In the IR, these models flow to Calabi–Yau sigma models, which the random couplings of the GLSM corresponding to random choices of complex structure moduli. Studying the disorder-averaged GLSMs then amounts to studying ensemble averages of Calabi–Yau sigma models.

The paper is an interesting contribution to the literature on disorder-averaged field theories, and the use of localisation to obtain exact results is particularly nice.

Some comments:

The disorder average is performed using a Gaussian distribution. It would be interesting to understand whether there is a real reason for doing this. For example, for Calabi–Yau sigma models, one might imagine that the natural way to average over the complex structure moduli space is to use the Weyl–Peterson measure on the moduli space. (A similar approach has been taken when considering disorder-averaged theories on the torus.) This would amount to requiring that the random couplings J are distributed with respect to the measure implied by the Zamolodchikov metric. This would clearly complicate the analysis, but may eventually lead to simple (and canonical) results.

It would also be interesting to see further work on GLSMs, particularly on understanding whether quantities such as Yukawa couplings have simple, disorder-averaged expressions which might help in understanding general features about both Calabi-Yau manifolds and, for phenomenology, the resulting particle physics models.

Requested changes

  1. It would be sensible to cite 2103.15826 for results on averaging over the complex structure moduli space of the torus, and its discussion in Section 8 on averaging Calabi-Yau CFTs.
  2. On page 32, below Equation (3.12), the authors write “\Delta_{1} and \Delta_{1}”, when I believe they mean “\Delta_{1} and \Delta_{2}”.

  • validity: high
  • significance: good
  • originality: high
  • clarity: top
  • formatting: excellent
  • grammar: excellent

Author:  Xiaoyang Shen  on 2024-04-22  [id 4439]

(in reply to Report 1 on 2024-03-09)

We thank the referee for the comments and have resubmitted the paper with the following adjustments:

1. It is more natural to calculate the Zamolodchikov metric and then do the subsequent averaging according to the measure. We included a discussion on averaging using the Weil–Petersson metric and cited 2103.15826 on page 33.
2. In the updated version, we have fixed the typos in eq 2.55 and on page 32.

---

## Round 1 · Referee Report · Anonymous (Referee 2) · 2024-3-12

Report

This paper discusses ensemble-averaging 2d SUSY CFTs, generalizing the Murugan-Stanford-Witten model. The standard methods used in ensemble averaged theories allows the authors to compute 2-point functions, 4-point functions and to extract the chaos exponent while SUSY also allows for some exact computations using localization.

The paper slightly lacks motivation. The original motivation for studying ensemble-averaged theories in this specific context was the large chaos exponent of the SYK model which hinted at a holographic dual, but the theories at hand never approach this regime. The appearance of a conformal manifold in the ensemble-averaged theory is also not new, with a non-SUSY example appearing already in [Gross,Rosenhaus].

The computations themselves are clear, exhaustive and precise. The comparison to exact SUSY results using localization is very convincing.

Requested changes

  1. small typos: (a) in eq 2.55, in the second equation it should be G_{\Phi^{(1)}} -> G_{\Phi^{(2)}} (b) on page 32, \Delta_1 and \Delta_1 -> \Delta_1 and \Delta_2

  2. It is not completely clear why solutions for the 2-point function become non-unitary for some values of \lambda. Taking section 2.4.1 as a specific example, there doesn't seem to be anything wrong with the model 2.76 for \lambda<1/2 before the ensemble average. Is the claim that there is no CFT for \lambda<1/2, or that it is a CFT but the ansatz 2.80 is wrong? For example for \lambda=0 the ansatz is probably wrong since the R-charge assignments would be found using just the first term in 2.76 and by c-maximization, and so the ansatz fails.

It is also not clear why the appearance of a moduli space at \lambda=1/2 leads to this non-unitarity. Is the claim that the moduli space leads to instabilities at \lambda<1/2?

  • validity: -
  • significance: -
  • originality: -
  • clarity: -
  • formatting: -
  • grammar: -

Author:  Xiaoyang Shen  on 2024-04-22  [id 4438]

(in reply to Report 2 on 2024-03-12)

We thank the referee for the comments and have resubmitted the paper with the following adjustments:

1. We thank the referee for pointing out the typos and have fixed the corresponding typos.
2. (a) The unitary bound of lambda arises due to the constraint that the coefficients of the two-point function should be positive. Another clue for the non-unitarity is that part of the spectrum read off from the four-point function becomes complex when lambda is smaller than 1/2. As the referee pointed out, the CFT before averaging should be unitary, and it remains a puzzle to us how disorder averaging generates non-unitarity. Another instance of this phenomenon is in the disordered vector model studied in 2112.09157. As discussed above figure 3, the spectrum becomes complex in some range of the parameter, while the theory before averaging does not exhibit any non-unitarity.
(b) When the lambda = 1/2, the theory becomes non-compact and has an emergent U(1) symmetry. Correspondingly we find a conserved spin-1 current in the spectrum, which indicates the flat direction in the supermultiplet.

---

## Editorial Decision

resubmitted